# Mathematical analysis of singularities in the diffusion model under the submanifold assumption

## Abstract

This paper provides several mathematical analyses of the diffusion model in machine learning. The drift term of the backward sampling process is represented as a conditional expectation involving the data distribution and the forward diffusion. The training process aims to find such a drift function by minimizing the mean-squared residue related to the conditional expectation. Using small-time approximations of the Green's function of the forward diffusion, we show that the analytical mean drift function in DDPM and the score function in SGM asymptotically blow up in the final stages of the sampling process for singular data distributions such as those concentrated on lower-dimensional manifolds, and is therefore difficult to approximate by a network. To overcome this difficulty, we derive a new target function and associated loss, which remains bounded even for singular data distributions. We validate the theoretical findings with several numerical examples.

## 1 Introduction

The field of generative models has emerged as a powerful tool for building probabilistic models and generating new samples from a given dataset. By accounting for the joint distribution of observable and target variables, these models offer a flexible and efficient way to analyze complex data. Generative models have been applied across a wide range of disciplines, including computer vision (Elasri et al., 2022), speech signal processing (Wali et al., 2022), natural language processing (Iqbal & Qureshi, 2022), and natural sciences (Strokach & Kim, 2022). Recent advances in generative models, including the popular variational autoencoder (VAE) (Kingma & Welling, 2014), generative adversarial network (GAN) (Goodfellow et al., 2014), flow-based model(Papamakarios et al., 2021), and DeepParticle model (Wang et al., 2022), have demonstrated their ability to solve diverse problems across different domains. These models share a common feature: they use neural network approximation to map an easy-to-sample distribution to an unknown distribution driven by data.

In contrast to these direct constructions, another type of generative model links distributions through one-parameter continuous deformations. This approach has a long history in the mathematical literature and involves solving the Langevin equation for sufficiently long times, starting from any distribution, to generate a standard normal distribution. For some distributions $p_{data}$, one may find a pair of functions $(v, D)$ such that the Fokker-Planck equation,

$$\rho_t = -\nabla \cdot (v\rho) + \nabla \cdot (\nabla(D\rho)) \quad (t, x) \in [0, 1] \times \mathbb{R}^d, \tag{1}$$

continuously deforms the easy-to-sample distribution $p_0 = \rho(0, \cdot)$ to $p_{data} = \rho(1, \cdot)$. Sampling from $X_0 \sim p_0$ and solving the SDE in $[0, 1]$,

$$dX_t = v(t, X_t)dt + \sqrt{2D(t, X_t)}dW_t, \tag{2}$$

we can generate $p_{data}$ by realization of $X_1$. A number of constructive approaches to $(v, D)$ have been considered in the literature, e.g. Song & Ermon (2019); Block et al. (2020); Wang et al. (2021). A celebrated one is the MCMC sampler, which samples from $p_{data} = \frac{1}{Z}\exp(-V(x))$ with $V$ typically arising from a log-likelihood function (Parisi, 1981). Then, starting from any reasonable initial distribution, $p_{data}$ is generated

by solving Eq. (2) with $(v, D) = (-\nabla V, I)$ over long times. Another example is the neural ordinary differential equation (NeuralODE) (Chen et al., 2018) which finds a pair $(v, D)$ from data without a closed-form representation of $V$. This continuous-time flow-based model connects the base distribution and the data distribution using an ordinary differential equation.

All the models mentioned above create a transformation between a base distribution and the data distribution. Once a model of $(v, D)$ is derived either analytically or from data, the stochastic differential equation (SDE) integrator can be used to numerically solve the SDE from initial to terminal time, thereby interpreting it as a generative model.

The diffusion model is a novel probabilistic generative model that converts white noise into a desired data distribution by learning an implicit transformation (Austin et al., 2021). Inspired by non-equilibrium thermodynamics (Sohl-Dickstein et al., 2015), Ho et al. (2020) proposed denoising diffusion probabilistic models (DDPMs), a class of latent variable models, as an early diffusion model. Later, Song et al. (2021) unified several earlier models through the lens of stochastic differential equations and proposed score-based generative models (SGMs). The backward process (generation of new samples) can be interpreted as solving Eq. (2) with a *tweak* that reverts the notation of time and the initial distribution after the tweak is assumed to be a standard normal distribution. Inspired by Anderson (1982), Song et al. (2021) proposed a specific set of $(v, D)$ from the related Fokker-Planck equation and derived a loss functional based on the forward process. (Luo, 2022) and (Yang et al., 2022) provide literature reviews from different perspectives.

Despite its success, the sampling process for diffusion models is extremely slow and the computational cost is high. In DDPMs (Ho et al., 2020), for instance, 1000 steps are typically needed to generate samples. Several works have attempted to accelerate the sampling process (Lu et al., 2022a;b; Zhang & Chen, 2022; Salimans & Ho, 2022; Jolicoeur-Martineau et al., 2021; Zhang et al., 2023). In addition, Zhang & Chen (2022) pointed out that there were dramatically different performances in terms of discretization error and training error when they trained the score function of SGM on different datasets. Karras et al. (2022) proposed to learn a denoiser and showed the relationship between the denoiser and the score function. This denoiser is similar to our proposed function CEM (see below) for empirical distributions. However, the asymptotic behavior of denoiser, as well as one in CEM, is not trivial for general distributions. It was only observed in numerical experiments while not rigorously proved in theory. Our current work aims to provide a mathematical analysis for such observations.

**Main Contributions of this work:** (1) We rigorously characterize the singularity of the score function when the target distribution is defined on an embedded manifold. This justifies the singularities observed in numerical experiments in Zhang & Chen (2022), and mentioned in Chen et al. (2023c;a); Lee et al. (2022); Bortoli (2022); Chen et al. (2023b); Nachmani et al. (2021). (2) We show the proposed parameterization of score function (CEM) bypass the problem of modeling a singular function and significantly improve the efficiency of the training process. (3) We configure the CEM, over the training schedule and the weight normalization, with respect to the asymptotic analysis of the singularities.

**Discussion of Related Work** Several recent works focus on the theoretical perspective of diffusion models. (Bortoli et al., 2021) formulate diffusion models as diffusion Schrödinger bridges, and then provide a convergence result by assuming the score estimate is bounded in $L^\infty$. (Lee et al., 2022) prove that SGMs have a rate of polynomial convergence by assuming the score estimate is bounded in $L^2$ and requiring a strong regularity assumption on the data distribution, i.e., log-Sobolev inequality. (Chen et al., 2023c) and (Chen et al., 2023a) relax these assumptions to obtain more general convergence results. However, they still require that the score function can be estimated well (in the sense of $L^2$).

The works mentioned above assume the score function can be estimated well by a neural network. However, this estimate will usually fail when dealing with singular data distributions, such as the distribution supported on a lower-dimensional manifold. (Bortoli, 2022) and (Zhang & Chen, 2022) explore the affect of singular data distributions, and they realize that the score function blows up as $t$ tends to zero for the singular data. Observing the blow-up behavior experimentally, the derivation in Bortoli (2022) is based on the assumed existence of blow-up as $t$ tends to zero. Especially, their theory does not prove such existence. In contrast, our theory provides mathematically rigorous analyses of asymptotic behavior at the final time, which includes both sharp upper and lower bound. More precisely, we find that the singular behavior of the

score function is $\frac{1}{t}$, which is a sharper result than Pidstrigach (2022) (they prove that $\|\nabla \log(p_t)\| \gtrsim \frac{1}{\sqrt{t}}$ for the Brownian motion example). Moreover, we notice a parallel work Chen et al. (2023b) which studies the score approximation under singular data distribution by requiring a linear assumption. In comparison, our work does not require such an assumption.

From the experimental perspective of diffusion models, DDPM and SGM conduct different weighting function for training the neural network to estimate the score function. (Karras et al., 2022; Salimans & Ho, 2022) and (Campbell et al., 2022) propose similar loss functions or training objectives for improving the performance of diffusion models. Despite these similarities in configurations, we provide a solid theoretical foundation for these configurations from rigorous mathematical analyses.

## 2 Background

As a class of probabilistic generative models, diffusion models are used to sample from a $d$-dimensional target probability distribution. Given a target distribution $p_{data}$ and a random variable $X_0 \sim p_{data}$, the general idea of diffusion models is to add noise to $X_0$ step by step such that $X_T$ is an easy-to-sample standard normal distribution. Subsequently, a reverse diffusion process is used to sample from $p_{data}$. The generative task of diffusion models is done by solving an SDE for the form (2) defined backward in time from $T$ to 0. In general, there is no closed-form expression for the reversion and we usually learn it from available data.

### 2.1 DDPM

As a main class of diffusion models, DDPMs (Ho et al., 2020) learn a distribution $p_\theta$ that approximates $p_{data}$ as follows. We start with the forward process, denoted by $X_k$ and $X_0 \sim p_{data}$. We gradually add Gaussian noise to the data with a schedule of $K$ steps at $\beta_1, \ldots, \beta_K$:

$$p(X_{1:K}|X_0) := \prod_{k=1}^{K} p(X_k|X_{k-1}) \tag{3}$$

and

$$p(X_k|X_{k-1}) := \mathcal{N}(\sqrt{1-\beta_k}X_{k-1}, \beta_k I_d). \tag{4}$$

Denoting $\alpha_k := 1 - \beta_k$ and $\bar{\alpha}_k := \prod_{s=1}^{k} \alpha_s$, we may recast it as:

$$\begin{aligned} X_{k+1} &= \sqrt{1-\beta_k}X_k + \sqrt{\beta_k}\epsilon, \\ &= \sqrt{\alpha_k}X_k + \sqrt{1-\alpha_k}\epsilon, \quad \epsilon \sim \mathcal{N}(0, I_d). \end{aligned} \tag{5}$$

A notable property of the forward diffusion process is that

$$p(X_k|X_0) = \mathcal{N}(\sqrt{\bar{\alpha}_k}X_0, (1-\bar{\alpha}_k)I_d), \tag{6}$$

which implies that the data is converted to a standard Gaussian distribution as $\bar{\alpha}_k$ converges to 0.

For the generative sampling task, we construct a backward process, denoted by $\widetilde{X_{0:K}}$, such that $\forall k$, $\widetilde{X_k}$ shares the same marginal distribution as $X_k$. As a starting point, $\widetilde{X_K}$ follows $\mathcal{N}(0, I_d)$, which is easy to sample. Then we iteratively find the conditional distribution

$$p_\theta(\widetilde{X_{k-1}}|\widetilde{X_k}) := \mathcal{N}(\widetilde{X_{k-1}}; \mu_\theta(\widetilde{X_k}, k), \Sigma_\theta(\widetilde{X_k}, k)), \tag{7}$$

where $(\mu_\theta, \Sigma_\theta)$ can be learned from data evolving according to the forward diffusion process (3).

Ho et al. (2020) proposed $\Sigma_\theta(\widetilde{X_k}, k) = \beta_k I_d$ and

$$\mu_\theta(\widetilde{X_k}, k) = \frac{1}{\sqrt{\alpha_k}}\Big(\widetilde{X_k} - \frac{\beta_k}{\sqrt{1-\bar{\alpha}_k}}\epsilon_\theta(\widetilde{X_k}, k)\Big), \tag{8}$$

where $\epsilon_\theta$ is modeled by a neural network. We then obtain samples from the distribution $p_\theta(\widetilde{X_{k-1}}|\widetilde{X_k})$ by computing

$$\widetilde{X_{k-1}} = \frac{1}{\sqrt{\alpha_k}}\left(\widetilde{X_k} - \frac{\beta_k}{\sqrt{1-\bar{\alpha}_k}}\epsilon_\theta(\widetilde{X_k},k)\right) + \sqrt{\beta_k}\widetilde{N_k}, \tag{9}$$

where $\widetilde{N_k} \sim \mathcal{N}(0, I_d)$.

For the reverse diffusion process, Ho et al. (2020) proposed finding the best trainable parameters $\theta$ by optimizing the variational lower bound:

$$L := E_p[-\log \frac{p_\theta(X_{0:K})}{p(X_{1:K}|X_0)}]. \tag{10}$$

They also found a simplified loss function, which improves sample quality:

$$L_{simple}(\theta) := E_{k,X_0,\epsilon}\|\epsilon - \epsilon_\theta(\sqrt{\bar{\alpha}_k}X_0 + \sqrt{1-\bar{\alpha}_k}\epsilon, k)\|^2. \tag{11}$$

## 2.2 Score-based generative model (SGM)

DDPM may be viewed as an SGM inferred from discretizations of stochastic differential equations (SDEs) (Song et al., 2021). The general idea of an SGM is to transform a data distribution into a known base distribution by means of an SDE, while the reverse-time SDE is used to transform the base distribution back to the data distribution. The forward process can be written as follows:

$$dX_t = h(X_t,t)dt + g(t)dW_t, \qquad X_0 \sim p_{data}, \tag{12}$$

where $W_t$ is a Brownian motion and $X_T \sim p_T$ approximates the standard normal distribution for large value of time $T$. The corresponding reverse-time SDE $\widetilde{X_t}$ shares the same marginal distribution as the forward process $X_t$ and hence gives a pair of $(v, D)$ in Eq. (1) and Eq. (2) modulo the reversing of the direction of time. It can be written as

$$d\widetilde{X_t} = \left[h(\widetilde{X_t},t) - g(t)^2\nabla_{\widetilde{X_t}}\log p_t(\widetilde{X_t})\right]dt + g(t)d\widetilde{W_t}, \tag{13}$$

where $p_t$ is the solution of the Fokker-Planck equation for the forward SDE (12), see Anderson (1982). Song et al. (2021) proposed learning the *score* function $\nabla_X \log p_t(X)$ by minimizing the score-matching loss function:

$$E_{t,X_0,X_t}\left[\lambda(t)\|S_\theta(X_t,t) - \nabla_{X_t}\log p_t(X_t|X_0)\|^2\right], \tag{14}$$

where $S_\theta(X_t,t)$ is a time-dependent score-based model, and $\lambda(t)$ is a positive weighting function.

## 2.3 Training and sampling issues for diffusion models

Diffusion models consist of two processes: a forward process and a reverse-time process. The forward process is used to transform the target distribution into a normal distribution. The forward process is explicitly given and does not require training. In contrast, the reverse-time process is used to restore the target distribution from the normal distribution, with coefficients that are not explicitly known and may be approximated by training.

Training of DDPM seeks $\theta$ by minimizing Eq. (11). For SGM, the evaluation of the score function $\nabla_X \log p_t(X)$ is not explicit. Song et al. (2021) discuss several approximations including Gaussian transition kernel $p_t(X_t|X_0)$ or using sliced score matching. In the next section, we provide an expression (23) as the loss function arising from the continuous solution of the Fokker-Planck equation.

In addition to the issue of accessibility of $\nabla_X \log p_t(X)$, the score function $\nabla_X \log p_t(X)$ in Eq. (14) may be complex and exhibit local structures and singularities, in particular near $t = 0$. This was pointed out by Dockhorn et al. (2022); Song et al. (2021); Zhang & Chen (2022), where they proposed experimental ways to deal with the singularity of the score function $\nabla_X \log p_t(X)$.

In the next section, we will elaborate on accessibility and regularity issues from a more mathematical point of view.

## 3 Generative diffusion model from a mathematical perspective

In this section, we first provide a unified framework for DDPM and SGM based on considering the backward-generating process as an SDE solver. The training schedule is interpreted as the discretization time step of the SDE solver. By exploring the analytical solution of the Fokker-Planck equation, we represent the score function as a natural conditional expectation. The training process then aims to find a network-based model that approximates the conditional expectation by minimizing the mean-squared prediction error function. In Section 3.2, we theoretically show that for the preceding generative models (SGM, DDPM), the conditional expectation exhibit singularities at $t \to 0$ under general sub-manifold assumption. The theoretical value of the model turns to infinity as the backward process approaches the terminal of the schedule. To overcome this difficulty, we finally propose a new model (CEM) that is bounded by the range of $p_{data}$.

Before starting the derivation, we would remark that a few works have proposed similar frameworks as in Section 2, such as Gong & Li (2021); Song et al. (2021); Dockhorn et al. (2022); Bao et al. (2022); Bortoli (2022) (about unified time). Meanwhile to address the stability, Karras et al. (2022) proposing to use a neural network to learn a denoiser ($X_0$ prediction), which is similar to our proposed function CEM of 3.3 in case of point cloud distributions. While the advantage of proposing such denoiser is only through experiment in Karras et al. (2022). (Salimans & Ho, 2022) and (Campbell et al., 2022) also propose similar loss functions or training objectives for improving the performance of diffusion models. In contrast, this section aims at presenting the theoretical reasoning that conventional models (e.g. DDPM, SGM) will suffer from instabilities during training for a large class of dataset (data that has locally low dimensional structures) regardless of configuration; while CEM or similar models avoid such singularities during training process. Hence we keep the derivation of the framework for completeness and generality of the theoretical results.

### 3.1 Unifying the time framework in DDPM and SGM

Inspired by SGM, the forward process of diffusion models may be viewed as a discretization of the following $d-$dimensional OU process:

$$dX_t = -\frac{1}{2}X_t dt + dW_t. \tag{15}$$

Consider a $K$ partition of the time interval $[0, T]$, $t_0 = 0 < t_1 < t_2 < \cdots < t_K = T$, where $t_k - t_{k-1} = \Delta t_k = -\log(1 - \beta_k)$ (equivalently $\beta_k = 1 - e^{-\Delta t_k}$). Then at time step $t_k$,

$$X_{t_{k+1}} = X_{t_k} e^{-\frac{\Delta t_k}{2}} + \int_{t_k}^{t_{k+1}} e^{-\frac{t_{k+1}-s}{2}} dW_s$$
$$\sim X_{t_k}\sqrt{1 - \beta_k} + \sqrt{\beta_k} N_k, \quad N_k \sim \mathcal{N}(0, I_d), \tag{16}$$

which coincides with the forward diffusion process in the diffusion models literature, e.g., Ho et al. (2020); Song et al. (2021). Compared to the preceding section, note that $\alpha_k = e^{-\Delta t_k}$, $\bar{\alpha}_k = e^{-t_k}$.

The advantage of the continuous model is twofold. First, we can represent the distribution of Eq. (15) at time $t$ by Eq. (17)

$$X_t \sim X_0 e^{-\frac{t}{2}} + (1 - e^{-t})N, \quad \text{where} \quad N \sim \mathcal{N}(0, I_d). \tag{17}$$

Second, we can easily estimate the time necessary to convert the real data distribution to normal distribution. Empirically, we take the final time $T > 10$ which leads to the fraction of the initial data in $X_T$ to be less than $\exp(-5) = 0.0067$.

The backward (sampling) process follows the reverse-time SDE (Anderson, 1982),

$$d\widetilde{X_t} = -(\frac{1}{2}\widetilde{X_t} + \nabla_X \log p(\widetilde{X_t}, t))dt + d\widetilde{W_t}, \tag{18}$$

where $p_t$ is a forward Kolmogorov equation of Eq. (15) with initial data distribution $p_{data}$ and $\widetilde{W_t}$ is a standard Brownian motion independent of $W_t$. Then $X_t$ and $\widetilde{X_t}$ share the same marginal distribution.

A key observation is that $p$, the law of the OU process, has an analytical solution (Evans, 2010),

$$p(X, t) = \frac{1}{Z} \int \exp\left( - \frac{\|X - X_0 e^{-\frac{t}{2}}\|^2}{2(1 - e^{-t})} \right) p_{data}(X_0) dX_0, \tag{19}$$

where $Z$ is a normalizing factor that depends on $t$ and $d$. The global score function $S$ may then be interpreted as a conditional expectation, namely,

$$
\begin{aligned}
S(X, t) &= -\nabla_X \log p(X, t) = -\frac{\nabla_X p(X, t)}{p(X, t)} \\
&= \frac{\frac{1}{Z} \int \frac{X - X_0 e^{-t/2}}{1 - e^{-t}} \exp\left( - \frac{\|X - X_0 e^{-t/2}\|^2}{2(1 - e^{-t})} \right) p_{data}(X_0) dX_0,}{\frac{1}{Z} \int \exp\left( - \frac{\|X - X_0 e^{-t/2}\|^2}{2(1 - e^{-t})} \right) p_{data}(X_0) dX_0} \\
&= E_{X_0}\left[ \frac{X_t - X_0 e^{-t/2}}{1 - e^{-t}} | X_t = X \right],
\end{aligned}
\tag{20}
$$

where $X_t$ follows the forward process (15). The conditional expectation in (20) can be interpreted as follows. Starting from $X_0$ following $p_{data}$, the forward process solving (15) gives the base distribution $X_t$. Given the observation of $X_t$ as $X$, $\frac{X_t - X_0 e^{-t/2}}{1 - e^{-t}}$ follows a posterior distribution. Taking the expectation of the posterior gives the analytical expression of $S$.

*Remark* 3.1. By standard Markov property, for $t' < t$,

$$S(X, t) = E_{X_{t'}}\left[ \frac{X_t - X_{t'} e^{-(t-t')/2}}{1 - e^{-(t-t')}} | X_t = X \right]. \tag{21}$$

**Training** In general, $S(x, t) : \mathbb{R}^d \times [0, T] \to \mathbb{R}^d$ is a very high dimensional function and hence lacks global approximation. Leveraging the properties of the conditional expectation for fixed $t$, $S(X, t)$ is the optimizer of the following mean-squared prediction error functional,

$$J(S) = E_{X_0, X_t}\left\| \frac{X_t - X_0 e^{-t/2}}{1 - e^{-t}} - S_\theta(X_t, t) \right\|^2. \tag{22}$$

By assigning a weight for the $t$-variable, the training process of Song et al. (2021) is generalized as seeking a network-represented function $S_\theta$ that minimizes,

$$
\begin{aligned}
& E_{X_0, N \sim \mathcal{N}(0, I_d), t} \\
& \left[ \lambda(t) \left\| \frac{N}{\sqrt{1 - e^{-t}}} - S_\theta(X_0 e^{-t/2} + \sqrt{1 - e^{-t}} N, t) \right\|^2 \right].
\end{aligned}
\tag{23}
$$

We remark that we use samples of $(X_0, t, N)$ for the evaluation of the integral in Eq. (23) and samples of $t$ do not necessarily follow the same schedule as those of the backward process.

**Sampling** There is no general closed-form solution for the backward process (18) and so we employ splitting schemes,

$$
\begin{cases}
\overline{X_{t_{k+1}}} = \widetilde{X_{t_{k+1}}} - \Delta t_k S_\theta(\widetilde{X_{t_{k+1}}}, t_{k+1}) \\
\widetilde{X_{t_k}} = e^{\Delta t_k/2} \overline{X_{t_{k+1}}} + \sqrt{1 - e^{-\Delta t_k}} \widetilde{N}_k
\end{cases}
\tag{24}
$$

where $\widetilde{N}_k \sim \mathcal{N}(0, I_d)$.

*Remark* 3.2. The training and sampling process exactly coincides with the aforementioned SGM, i.e., learning the score function $S(x, t)$ with $L^2$-norm. It is also related to DDPM as follows. Taking $\epsilon_\theta(x, t) = \sqrt{1 - e^{-t}} S_\theta(x, t)$, the loss functions (23) becomes,

$$
\begin{aligned}
& \int_t \frac{\lambda(t)}{1 - e^{-t}} E_{X_0, N \sim \mathcal{N}(0, I_d)} \left\| N - \epsilon_\theta(X_0 e^{-t/2} + \sqrt{1 - e^{-t}} N, t) \right\|^2 dt \\
& \approx \sum_k \frac{\lambda(t_k)}{1 - \bar{\alpha}_k} E_{X_0, N \sim \mathcal{N}(0, I_d)} \left\| N - \epsilon_\theta(X_0 \sqrt{\bar{\alpha}_k} + \sqrt{1 - \bar{\alpha}_k} N, t_k) \right\|^2.
\end{aligned}
\tag{25}
$$

And the sampling process (24),

$$\widetilde{X_{t_k}} = e^{\Delta t_k/2}(\widetilde{X_{t_{k+1}}} - \Delta t_k S_\theta(\widetilde{X_{t_{k+1}}}, t_{k+1})) + \sqrt{1 - e^{-\Delta t_k}}\widetilde{N}_k \tag{26}$$

$$= \frac{1}{\sqrt{\alpha_k}}(\widetilde{X_{t_{k+1}}} - \frac{\Delta t_k}{\sqrt{1 - \bar{\alpha}_k}}\epsilon_\theta(\widetilde{X_{t_{k+1}}}, t_{k+1})) + \sqrt{\beta_k}\widetilde{N}_k$$

$$\approx \frac{1}{\sqrt{\alpha_k}}(\widetilde{X_{t_{k+1}}} - \frac{1 - \alpha_k}{\sqrt{1 - \bar{\alpha}_k}}\epsilon_\theta(\widetilde{X_{t_{k+1}}}, t_{k+1})) + \sqrt{\beta_k}\widetilde{N}_k. \tag{27}$$

For the approximation in the last line, we use $1 - \alpha_k = \beta_k \approx \Delta t_k$. This scheme coincides with DDPM using $L_{simple}$ loss function Eq. (11).

### 3.2 Singularity of the score function

In the previous subsection, we showed the conventional training process aimed to approximate the conditional expectation function $S(X, t) = E_{X_0}[\frac{X_t - X_0 e^{-t/2}}{1 - e^{-t}}|X_t = X]$ or $\epsilon(X, t) = \sqrt{1 - e^{-t}}S(X, t)$. However, such functions potentially exhibit singularities near $t = 0$, which corresponds to the last few steps of the sampling process. For example, if $X_0$ follows a single point distribution, then $S(X, t) = \frac{X - X_0 e^{-t/2}}{1 - e^{-t}}$ while $\epsilon(X, t) = \frac{X - X_0 e^{-t/2}}{\sqrt{1 - e^{-t}}}$. It is then very difficult for general network propagation configurations to express such a blow-up as $t \to 0$.

An $n$-dimensional sub-manifold is denoted by $\Omega$, where $\Omega \subset \mathbb{R}^d$ and $n < d$. To characterize such asymptotics for most general datasets, we made the following assumptions over point $X$ in the backward (sampling) process and data distribution $p_{data}$.

**(H1) Uniqueness Assumption** Fixing point $X$, we denote the $y_X$ on $\Omega$ as the unique point that minimize the distant between $X$ and $\Omega$, i.e. $y_X = \arg\min_{y \in \Omega} \|y - X\|$ is uniquely defined.

**(H2) Subspace Assumption** Let $B_\varepsilon = \{y \in \Omega : \|y - X\| < \|y_X - X\| + \varepsilon\}$, which is decreasing set series as $\varepsilon \to 0$. We assume there exists $0 < \varepsilon_0 \ll 1$, such that for $y \in B_{\varepsilon_0}$, there exists a local coordinate chart, $z \to y(z) \in B_{\varepsilon_0} \subset \Omega$, under which $p_{data}$ is assumed to have a locally defined smooth density function in form of,

$$p_{data}(y) = \hat{\rho}(z)|J(z)|\delta_{y(z) \in \Omega}, \tag{28}$$

where $J$ is the Jacobian of local coordinate transformation and the size of $J$ is corresponding to the dimension of low-dimensional variable $z$, denoted as $n$. In addition, we assume within $y(z) \in B_{\varepsilon_0}$, $\hat{\rho}(z)$ is continuous and bounded,

$$0 < \rho_0 \le \hat{\rho}(z)|J(z)| \le \rho_1 < \infty. \tag{29}$$

Under these assumptions, we state the first key theorem of this work as follows.

**Theorem 3.3.** *(Singularity of the score functions) Let $X \in \mathbb{R}^d \backslash \Omega$ and data distribution $p_{data}$ satisfy (H1) and (H2). Then, the score function $S(X, t)$ blows up as $t \to 0$, and more precisely, satisfies*

$$S(X, t) = \frac{X - y_X}{t}(1 + o(1)). \tag{30}$$

*Proof.* The score function has the following representation

$$S(X, t) = E_{X_0}[\frac{X_t - X_0 e^{-\frac{t}{2}}}{1 - e^{-t}}|X_t = X] = \frac{g(X, t)}{1 - e^{-t}}, \tag{31}$$

$$g(X, t) = E_{X_0}[X - X_0 e^{-\frac{t}{2}}|X_t = X] = \frac{\int_\Omega (X - y e^{-\frac{t}{2}})e^{-\frac{\|X - y e^{-\frac{t}{2}}\|^2}{2(1 - e^{-t})}} p_{data}(y)dy}{\int_\Omega e^{-\frac{\|X - y e^{-\frac{t}{2}}\|^2}{2(1 - e^{-t})}} p_{data}(y)dy} \tag{32}$$

With a fixed $\varepsilon > 0$, we decompose $g$ into two parts,

$$g(X,t) = \frac{\int_{B_\varepsilon}(X - ye^{-\frac{t}{2}})e^{-\frac{\|X - ye^{-\frac{t}{2}}\|^2}{2(1-e^{-t})}}p_{data}(y)dy}{\int_\Omega e^{-\frac{\|X - ye^{-\frac{t}{2}}\|^2}{2(1-e^{-t})}}p_{data}(y)dy} + \frac{\int_{\Omega \backslash B_\varepsilon}(X - ye^{-\frac{t}{2}})e^{-\frac{\|X - ye^{-\frac{t}{2}}\|^2}{2(1-e^{-t})}}p_{data}(y)dy}{\int_\Omega e^{-\frac{\|X - ye^{-\frac{t}{2}}\|^2}{2(1-e^{-t})}}p_{data}(y)dy}. \tag{33}$$

By definition of $B_\varepsilon$, for $y \in \Omega \backslash B_\varepsilon$,

$$\|X - ye^{-\frac{t}{2}}\| \geq e^{-\frac{t}{2}}\|X - y\| - (1 - e^{-\frac{t}{2}})\|X\| \geq e^{-\frac{t}{2}}(\|X - y_X\| + \epsilon) - (1 - e^{-\frac{t}{2}})\|X\| =: C_{t,\varepsilon}. \tag{34}$$

For $y \in B_\varepsilon$,

$$\|X - ye^{-\frac{t}{2}}\| \leq e^{-\frac{t}{2}}\|X - y\| + (1 - e^{-\frac{t}{2}})\|X\| \leq e^{-\frac{t}{2}}(\|X - y_X\| + \epsilon) + (1 - e^{-\frac{t}{2}})\|X\| =: D_{t,\varepsilon}. \tag{35}$$

We claim that the second term of (33) converges to zero as $t \to 0$ (with fixed $\varepsilon$) since

$$\left\| \frac{\int_{\Omega \backslash B_\varepsilon}(X - ye^{-\frac{t}{2}})e^{-\frac{\|X - ye^{-\frac{t}{2}}\|^2}{2(1-e^{-t})}}p_{data}(y)dy}{\int_\Omega e^{-\frac{\|X - ye^{-\frac{t}{2}}\|^2}{2(1-e^{-t})}}p_{data}(y)dy} \right\| \leq \frac{\int_{\Omega \backslash B_\varepsilon}(\|X\| + \|y\|)e^{-\frac{C_{t,\epsilon}^2}{2(1-e^{-t})}}p_{data}(y)dy}{\int_\Omega e^{-\frac{\|X - ye^{-\frac{t}{2}}\|^2}{2(1-e^{-t})}}p_{data}(y)dy} \tag{36}$$

$$\leq \frac{\int_\Omega (\|X\| + \|y\|)p_{data}(y)dy}{\int_\Omega e^{-\frac{\|X - ye^{-\frac{t}{2}}\|^2 - C_{t,\epsilon}^2}{2(1-e^{-t})}}p_{data}(y)dy}. \tag{37}$$

Given the boundedness of the expectation of the data distribution $p_{data}$, it remains to show the denominator converges to infinity as $t \to 0$. In fact, with (35) in mind,

$$\int_\Omega e^{-\frac{\|X - ye^{-\frac{t}{2}}\|^2 - C_{t,\epsilon}^2}{2(1-e^{-t})}}p_{data}(y)dy \geq \int_{B_{\varepsilon'}} e^{-\frac{\|X - ye^{-\frac{t}{2}}\|^2 - C_{t,\epsilon}^2}{2(1-e^{-t})}}p_{data}(y)dy \geq \int_{y(z) \in B_{\varepsilon'}} e^{-\frac{D_{t,\varepsilon'}^2 - C_{t,\epsilon}^2}{2(1-e^{-t})}}\hat{\rho}(z)|J(z)|dz. \tag{38}$$

With $t$ sufficient small, say $t < t_0$ such that $\frac{\varepsilon}{2} > 2(e^{\frac{t_0}{2}} - 1)\|X\|$, we set $\varepsilon' = \frac{\varepsilon}{2} - 2(e^{\frac{t_0}{2}} - 1)\|X\| > 0$ so that $\forall\, 0 < t < t_0$,

$$C_{t,\varepsilon}^2 - D_{t,\varepsilon'}^2 = \left(e^{-\frac{t}{2}}(\varepsilon - \varepsilon') - 2(1 - e^{-\frac{t}{2}})\|X\|\right)e^{-\frac{t}{2}}(2\|X - y_X\| + \varepsilon + \varepsilon') \tag{39}$$

$$= \left(e^{-\frac{t}{2}}\left(\frac{\varepsilon}{2} + 2(e^{\frac{t_0}{2}} - 1)\|X\|\right) - 2(1 - e^{-\frac{t}{2}})\|X\|\right)e^{-\frac{t}{2}}(2\|X - y_X\| + \varepsilon + \varepsilon') \tag{40}$$

$$= \left(e^{-\frac{t}{2}}\frac{\varepsilon}{2} + 2(e^{\frac{t_0-t}{2}} - 1)\|X\|\right)e^{-\frac{t}{2}}(2\|X - y_X\| + \varepsilon + \varepsilon') \tag{41}$$

$$\geq e^{-t_0}\varepsilon\|X - y_X\| > 0. \tag{42}$$

The right-hand side of (38) converges to infinity as $t \to 0$.

Similarly,

$$\frac{\int_{\Omega \backslash B_\varepsilon} e^{-\frac{\|X - ye^{-\frac{t}{2}}\|^2}{2(1-e^{-t})}}p_{data}(y)dy}{\int_{B_\varepsilon} e^{-\frac{\|X - ye^{-\frac{t}{2}}\|^2}{2(1-e^{-t})}}p_{data}(y)dy} \leq \frac{\int_{\Omega \backslash B_\varepsilon} e^{-\frac{\|X - ye^{-\frac{t}{2}}\|^2}{2(1-e^{-t})}}p_{data}(y)dy}{\int_{B_{\varepsilon'}} e^{-\frac{\|X - ye^{-\frac{t}{2}}\|^2}{2(1-e^{-t})}}p_{data}(y)dy} \tag{43}$$

$$\leq \frac{\int_{\Omega \backslash B_\varepsilon} e^{-\frac{C_{t,\varepsilon}^2}{2(1-e^{-t})}}p_{data}(y)dy}{\int_{B_{\varepsilon'}} e^{-\frac{D_{t,\varepsilon'}^2}{2(1-e^{-t})}}p_{data}(y)dy} \leq \frac{1}{\int_{y(z) \in B_{\varepsilon'}} e^{-\frac{D_{t,\varepsilon'}^2 - C_{t,\varepsilon}^2}{2(1-e^{-t})}}\hat{\rho}(z)|J(z)|dz} = o(t) \tag{44}$$

So the denominator in the first term of (33) can also be decomposed and approximated by the contribution in $B_\varepsilon$

$$\int_\Omega e^{-\frac{\|X-ye^{-\frac{t}{2}}\|^2}{2(1-e^{-t})}} p_{data}(y)dy = (1+o(t)) \int_{B_\varepsilon} e^{-\frac{\|X-ye^{-\frac{t}{2}}\|^2}{2(1-e^{-t})}} p_{data}(y)dy. \tag{45}$$

Then when $t \to 0$, we have in local coordinates (28),

$$\frac{\int_{B_\varepsilon} (X - ye^{-\frac{t}{2}})e^{-\frac{\|X-ye^{-\frac{t}{2}}\|^2}{2(1-e^{-t})}} p_{data}(y)dy}{\int_{B_\varepsilon} e^{-\frac{\|X-ye^{-\frac{t}{2}}\|^2}{2(1-e^{-t})}} p_{data}(y)dy} \tag{46}$$

$$= \frac{\int_{y(z)\in B_\varepsilon} (X - y(z)e^{-\frac{t}{2}})e^{-\frac{\|X-y(z)e^{-\frac{t}{2}}\|^2}{2(1-e^{-t})}} \hat\rho(z)|J(z)|dz}{\int_{y(z)\in B_\varepsilon} e^{-\frac{\|X-y(z)e^{-\frac{t}{2}}\|^2}{2(1-e^{-t})}} \hat\rho(z)|J(z)|dz} \tag{47}$$

$$= X - e^{-\frac{t}{2}} \frac{\int_{y(z)\in B_\varepsilon} y(z)e^{-\frac{\|X-y(z)e^{-\frac{t}{2}}\|^2}{2(1-e^{-t})}} \hat\rho(z)|J(z)|dz}{\int_{y(z)\in B_\varepsilon} e^{-\frac{\|X-y(z)e^{-\frac{t}{2}}\|^2}{2(1-e^{-t})}} \hat\rho(z)|J(z)|dz} \tag{48}$$

Taking (29) into account, and realizing that $y(z)e^{-\frac{t}{2}}$ is well approximated by $y_X$ on $B_\epsilon$ for $t$ small,

$$\left\| \frac{\int_{y(z)\in B_\varepsilon} y(z)e^{-\frac{\|X-y(z)e^{-\frac{t}{2}}\|^2}{2(1-e^{-t})}} \hat\rho_t(z)|J(z)|dz}{\int_{y(z)\in B_\varepsilon} e^{-\frac{\|X-y(z)e^{-\frac{t}{2}}\|^2}{2(1-e^{-t})}} \hat\rho_t(z)|J(z)|dz} - y_X \right\| \tag{49}$$

$$\leq \frac{\rho_1 \int_{y(z)\in B_\varepsilon} \|y(z) - y_X\| e^{-\frac{\|X-y(z)e^{-\frac{t}{2}}\|^2}{2(1-e^{-t})}} dz}{\rho_0 \int_{y(z)\in B_\varepsilon} e^{-\frac{\|X-y(z)e^{-\frac{t}{2}}\|^2}{2(1-e^{-t})}} dz} \tag{50}$$

$$\leq \frac{\rho_1 \int_{y(z)\in B_\varepsilon} \varepsilon e^{-\frac{\|X-y(z)e^{-\frac{t}{2}}\|^2}{2(1-e^{-t})}} dz}{\rho_0 \int_{y(z)\in B_\varepsilon} e^{-\frac{\|X-y(z)e^{-\frac{t}{2}}\|^2}{2(1-e^{-t})}} dz} \tag{51}$$

$$\leq \frac{\varepsilon\rho_1}{\rho_0} \tag{52}$$

Substituting back to (48) then (32) we have

$$\lim_{t\to 0} g(X,t) = X - y_X + \mathcal{O}(\varepsilon). \tag{53}$$

Since choice of $\varepsilon > 0$ is arbitrary, from (31) we have,

$$\lim_{t\to 0} tS(X,t) = \lim_{t\to 0} \frac{t(X - y_X)}{1 - e^{-t}} = X - y_X. \tag{54}$$

$\square$

*Remark* 3.4. The above derivation may be generalized as an application of the Laplace method, which we now briefly present. The manifold $\Omega$ is covered by charts mapping subsets to domains of Euclidean space. Consider one such chart parameterized by variables $y = y(z)$ with $z \in U \subset \mathbb{R}^n$ and Jacobian of the transformation equals 1 to simplify. We assume that the closest point $y_X = y_X(z_X)$ for $z_X \in U$.

The Laplace method (see, e.g., Erdélyi (1956)) states that an integral of the form

$$\int_U e^{-\frac{1}{t}\theta(z)}h(z)dz \tag{55}$$

is approximated by,

$$(2\pi t)^{\frac{n}{2}}\frac{h(z_0)}{|H\theta(z_0)|^{\frac{1}{2}}}e^{-\frac{1}{t}\theta(z_0)}(1+o(1)) \tag{56}$$

where $z_0$ is the unique point minimizing $\theta(z)$ and $|H\theta(z_0)|$ is the determinant of the positive definite Hessian of $\theta$ at $z_0$.

We start with,

$$S(X,t) = \frac{\int_U (X-y(z)e^{-\frac{t}{2}})e^{-\frac{\|X-y(z)e^{-\frac{t}{2}}\|^2}{2(1-e^{-t})}}\hat{\rho}(z)dz}{(1-e^{-t})\int_U e^{-\frac{\|X-y(z)e^{-\frac{t}{2}}\|^2}{2(1-e^{-t})}}\hat{\rho}(z)dz}. \tag{57}$$

By noticing $1-e^{-t}\approx t$ when $t\to 0$ and applying the Laplace method (55) to the nominator and denominator with $\theta(z) = \|X-y(z)e^{-\frac{t}{2}}\|^2/2$ and $h(z) = (X-y(z)e^{-\frac{t}{2}})\hat{\rho}(z)$ and $h(z) = \hat{\rho}(z)$ correspondingly, we immediately arrive at,

$$S(X,t) = \frac{X-y_X}{t}(1+o(1)). \tag{58}$$

*Remark* 3.5. The uniqueness assumptions of Theorem 3.3 hold almost surely during the backward process. This is due to during the backward process, the target function (e.g. $S$ and $\epsilon$) is evaluated on an approximated distribution of the forward process, which is globally supported over $\mathbb{R}^d$ and hence almost surely outside of $\Omega$. As for the subspace assumption, It is a widely shared belief that data distribution in, e.g., human genes, climate patterns, and images, are supported on low dimensional structures (Tenenbaum et al., 2000; Roweis & Saul, 2000; Belkin & Niyogi, 2003). We would like to further remark that the dimension $n$ in the subspace assumption is only locally defined and our result holds as long as $n < d$.

In contrast to the above result, there are situations where the target functions ($\epsilon$ and $S$) remain bounded as $t$ approaches zero for specific distributions $p_{data}$.

**Theorem 3.6.** *(Regularity of the score function) Assuming the data distribution $X_0$ has the following form of probability density function,*

$$\rho = \rho_0 * \mu_1, \tag{59}$$

*where $\rho_0$ is some positive PDF and $\mu_1$ is PDF of normal distribution with variance $\sigma^2 > 0$. Then fixing $X$,*

$$-\nabla_X \log p(X,t) = E_{\hat{X}_0}\left[\frac{X_t - e^{-t/2}\hat{X}_0}{\sigma^2 e^{-t} + 1 - e^{-t}}|X_t = X\right], \tag{60}$$

*where $\hat{X}_0$ follows distribution whose PDF is $\rho_0$. The score function $-\nabla_X \log p(X,t)$ remains bounded when the support of $\rho_0$ is compact.*

*Proof.* Let $N_1$ and $N_2$ be independent standard normal distribution with variance $\sigma^2$ and $1-e^{-t}$ correspondingly. Denoting $\hat{X}_0$ as a random variable with distribution $\rho_0$ and $X_0$ is also a random variable with distribution $\rho$. Notice that $\rho = \rho_0 * \mu_1$, where $\mu_1$ is PDF of $N_1$. Therefore, we know that $X_0 = \hat{X}_0 + N_1$. Moreover, the solution of forward process (15) is $X_t = X_0 e^{-\frac{t}{2}} + \sqrt{1-e^{-t}}N$, where $N$ is a standard normal random variable. Subsequently, the equation $X_t = (\hat{X}_0 + N_1 e^{-\frac{t}{2}}) + N_2$ is hold in distribution sense. Using this relation, we can derive

$$E[e^{-t/2}X_0|X_t = X]$$

$$=E[e^{-t/2}(\hat{X}_0 + N_1)|e^{-t/2}(\hat{X}_0 + N_1) + N_2 = X]$$

$$=E[e^{-t/2}\hat{X}_0|e^{-t/2}(\hat{X}_0 + N_1) + N_2 = X] + E[E[e^{-t/2}N_1|e^{-t/2}N_1 + N_2 = X - e^{-t/2}\hat{X}_0, \hat{X}_0]]$$

$$=E[e^{-t/2}\hat{X}_0|e^{-t/2}(\hat{X}_0 + N_1) + N_2 = X] + E[\frac{\sigma^2 e^{-t}(X - e^{-t/2}\hat{X}_0)}{\sigma^2 e^{-t} + 1 - e^{-t}}|e^{-t/2}(\hat{X}_0 + N_1) + N_2 = X]$$

$$=E[\frac{\sigma^2 e^{-t}X + (1 - e^{-t})e^{-t/2}\hat{X}_0)}{\sigma^2 e^{-t} + 1 - e^{-t}}|e^{-t/2}(\hat{X}_0 + N_1) + N_2 = X]. \tag{61}$$

So

$$-\nabla_X \log p(X, t) = E_{X_0}[\frac{X_t - X_0 e^{-t/2}}{1 - e^{-t}}|X_t = X]$$

$$= E_{\hat{X}_0}[\frac{X_t - e^{-t/2}\hat{X}_0}{\sigma^2 e^{-t} + 1 - e^{-t}}|X_t = X], \tag{62}$$

which remain bounded when the support of $\rho_0$ is compact. $\qquad\square$

Theorem 3.6 provides a possible explanation for why samples of the DDPM and SGM seem to be randomly perturbed away from the possible local support of the data distribution manifold. As discussed in Theorem 3.3, the theoretical value of the target function becomes unbounded as $t$ approaches 0, which is not expressible by most network configurations. And the loss function that relies on the target function becomes unbounded too. The model turns out to learn a bounded function instead of a singular function, which corresponds to learning a polluted data distribution $\rho$ instead of $\rho_0$ ($p_{data}$). A sample from $\rho = \rho_0 * \mu_1$ can be viewed as adding independent Gaussian noise to a sample from the original distribution $p_{data}$.

Summarizing the above, forcing the network, upper bounded by $\frac{1}{\sigma^2}$, to learn the model $S$ or $\epsilon$ from data supported on a low-dimensional geometry, turns out to add i.i.d. Gaussian noise in each dimension with variance $\sigma^2$ to the original data.

### 3.3 A new model based on conditional expectation

To avoid such pollution, we propose the conditional expectation model (CEM) to respect the singularities. Note that,

$$S(X, t) = \frac{X}{1 - e^{-t}} - \frac{e^{-t/2}}{1 - e^{-t}} E_{X_0}[X_0|X_t = X]. \tag{63}$$

Denoting $E_{X_0}[X_0|X_t = X]$ as $f(X, t)$, we know for fixed $t$ that $f(\cdot, t)$ minimizes the following functional,

$$J(f) = E_{X_0}(\|X_0 - f(X_t, t)\|)^2. \tag{64}$$

This justifies defining a new loss function for training $f_\theta$ as

$$E_{X_0, X_t, t}\left[\lambda(t)\big\|X_0 - f_\theta(X_t, t)\big\|^2\right], \tag{65}$$

where $\lambda(t) > 0$ is a time-dependent weighing function that remains free for the user to choose.

*Remark* 3.7. A good choice of $\lambda$ is to align the training process for each $t$. While the analytical value is inaccessible without knowledge of data distribution, in practice, we employ $\lambda(t) = (e^t - 1)^{-1}$. This is inspired by an analysis of $L_{simple}$ in DDPM discussed as follows. In DDPM, the optimal

$$\epsilon_\theta(X, t) = E[\frac{X_t - X_0 e^{-t/2}}{\sqrt{1 - e^{-t}}}|X_t = X]. \tag{66}$$

Fixing $t$, the penalty function is lower bounded, i.e.,

$$E_{X_t}(\epsilon_\theta(X_t, t) - \frac{X_t - X_0 e^{-t/2}}{\sqrt{1 - e^{-t}}})^2 \geq E_{X_t} Var[\frac{X_t - X_0 e^{-t/2}}{\sqrt{1 - e^{-t}}} | X_t], \tag{67}$$

where the equality holds when (66) holds. Note that

$$Var[\frac{X_t - X_0 e^{-t/2}}{\sqrt{1 - e^{-t}}} | X_t] = \frac{e^{-t}}{1 - e^{-t}} Var(X_0 | X_t). \tag{68}$$

The uniform weight in $L_{simple}$ implies lower bound in the right-hand side of Eq. (67) is **assumed** to be independent of $t$. With the same assumption, we have

$$E_{X_t} Var(X_0 | X_t) \propto (e^t - 1), \tag{69}$$

which in turn gives $\lambda(t) = (e^t - 1)^{-1}$.

**Sampling process**  After training for $f_\theta$, the closed form solution of the backward process (18) remains unknown. Thus, we still need to use numerical SDE solvers to construct a generative model of $p_{data}$. Using (63), we consider the following replacement in the sampling scheme (24),

$$S_\theta(X, t) = \frac{X}{1 - e^{-t}} - \frac{e^{-t/2}}{1 - e^{-t}} f_\theta(X, t). \tag{70}$$

Even after re-directing the network to model a bounded function $f$, the drift term in the backward process, $-\frac{1}{2} X - \nabla_X \log p(X, t)$, may still be of order $\mathcal{O}(\frac{1}{t})$ near $t = 0$; see (30). The training schedule should be adapted accordingly. A natural choice is to match the drift scale with a single time step. At the time $t_k$ for $k > 1$, we consider the scale of changes due to the drift,

$$(t_k - t_{k-1})\frac{1}{t_k} := \gamma_k. \tag{71}$$

Minimizing $\gamma_k$ for all $k > 1$, we arrive at the following exponential schedule,

$$t_k = t_1 (1 - \gamma)^{1-k} \tag{72}$$

where $\gamma = 1 - (\frac{T}{t_1})^{\frac{1}{K-1}}$.

*Remark* 3.8. Though the scale of drift indicated in Eq. (30), i.e., $\mathcal{O}(\frac{1}{1 - e^{-t}})$, only applies when $t$ is near 0. For $t \gg 0$, we still use exponential schedule (72) to reduce the time of network evaluation.

## 4 Experiments

The experiments consist of five parts. First, we employ DDPM, SGM, and the proposed CEM to learn to generate a one-dimensional supported distribution in $\mathbb{R}^2$. By comparing the learned models with the corresponding analytic values, we show the new model outperforms DDPM and SGM by avoiding approximating singularities. Second, we also verify that if we replace the network with its corresponding analytic expression, the sampling process gives the exact distribution. Third, we investigate the performance of the new model depending on some parameters that are decided empirically. Fourth, we apply CEM to a high-dimensional example, i.e., the MNIST dataset, and compare the performance with DDPM. Lastly, we conduct ablation studies for the sampling schedule and the weighting function.

In all subsequent experiments, unless otherwise specified, we setup the diffusion model with $T = 10$ as the 'final' time and $K = 200$ uniform/non-uniform time grid points (exponential schedule (72)) for training/sampling. For model training, we use $10^6$ samples with a batch size of $10^4$, and we choose **Adam** as the optimizer, where the learning rate is 0.001. The network configuration will be specified in each example.

### 4.1 Comparison between SGM, DDPM and proposed CEM

In the following, we compare the SGM, DDPM, and CEM on several two-dimensional target distributions.

**Line normal distribution in 2d space**  As the first example, we consider a distribution supported on a line in two-dimensional space. Precisely, the data distribution is generated by $X = (X_1, X_2)$, where $X_1 \sim \mathcal{N}(0,1)$, $X_2 = 0$. In Appendix A.2, we derive the explicit formulas:

$$\begin{cases} S(X,t) = (X_1, \frac{X_2}{1-e^{-t}}), \\ \epsilon(X,t) = (\sqrt{1-e^{-t}}X_1, \frac{X_2}{\sqrt{1-e^{-t}}}), \\ f(X,t) = (X_1 e^{-\frac{t}{2}}, 0). \end{cases} \tag{73}$$

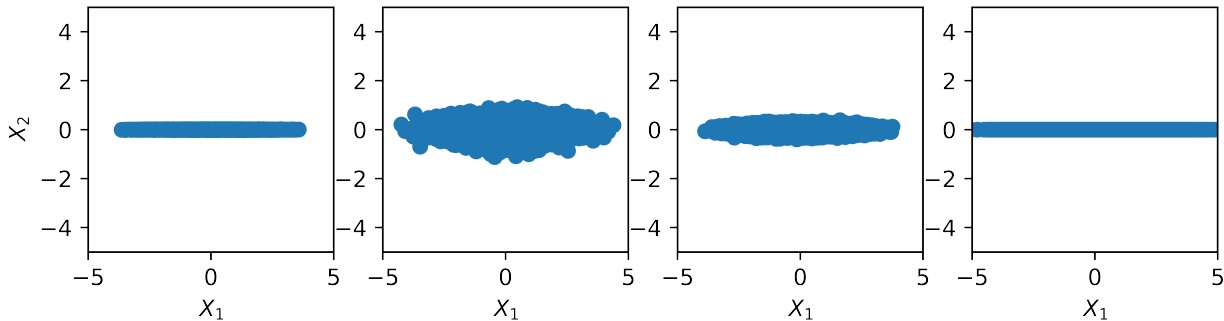

Figure 1: 1d line normal distribution in 2d space. From left to right: CEM, SGM, DDPM, and the ground truth. The network configuration is as follows: 2 hidden layers, each layer with 16 nodes, and **Tanh** as the activation function.

In Figure 1, we compare the distributions generated after training the CEM, SGM and DDPM. Our method CEM displays less pollution errors compared to DDPM and SGM. To verify that the errors originate from the poor approximation of the goal functions, we compare in Figure 2 the estimated score function $S_\theta(X,t)$ with the ground truth $S(X,t)$ at a fixed point $X_{eva} = (1, -0.1)$, i.e., $e(S(X_{eva},t), S_\theta(X_{eva},t)) = \|S(X_{eva},t) - S_\theta(X_{eva},t)\|$. We similarly define $e(\epsilon(X_{eva},t), \epsilon_\theta(X_{eva},t))$ and $e(f(X_{eva},t), f_\theta(X_{eva},t))$. Notice that $X_{eva}$ is outside of the support of the distribution $\mathbb{R} \times \{0\}$ and that by (73), the target score functions $S$ and $\epsilon$ exhibit singularities in the second coordinate. Correspondingly, in the left of Figure 2, we observe that the approximations of $f$, $S$, and $\epsilon$ are roughly correct for the first coordinate. This also verifies that the training of SGM and DDPM is indeed modeling the conditional expectation suggested in Eq. (20). On the right picture of Figure 2, we observe that, due to the existence of singularities, the approximations of $S$ and $\epsilon$ are incorrect and the error grows rapidly in the final steps of the sampling procedure.

Figure 3 displays the $L^2$-error between the analytic formulas in (73) and the estimated functions $f$, $S$ and $\epsilon$ obtained during the last 100 sampling steps in the backward process. The $L^2$ norm is defined as follows. With $S_\theta(X,t)$ the estimated score function, the $L^2$-error at a fixed time $t > 0$ is defined by

$$e_p(S, S_\theta) = \int \|S(X,t) - S_\theta(X,t)\|^2 p(X,t) dX, \tag{74}$$

where $p(X,t)$ is distribution of the forward process (15). In practice, we solve the forward process (15) to obtain the empirical distribution at time $t$ as an approximation of distribution $p$. We similarly evaluate $e_p(\epsilon, \epsilon_\theta)$ and $e_p(f, f_\theta)$.

Since $\epsilon = \sqrt{1-e^{-t}}S$ is of order $\mathcal{O}(\frac{1}{\sqrt{t}})$, we remark that with same configuration of network, $\epsilon$ in DDPM is better approximated than $S$ in SGM (see Figure 2 and Figure 3). This results in less pollution in the sampling process, as shown in Figure 1.

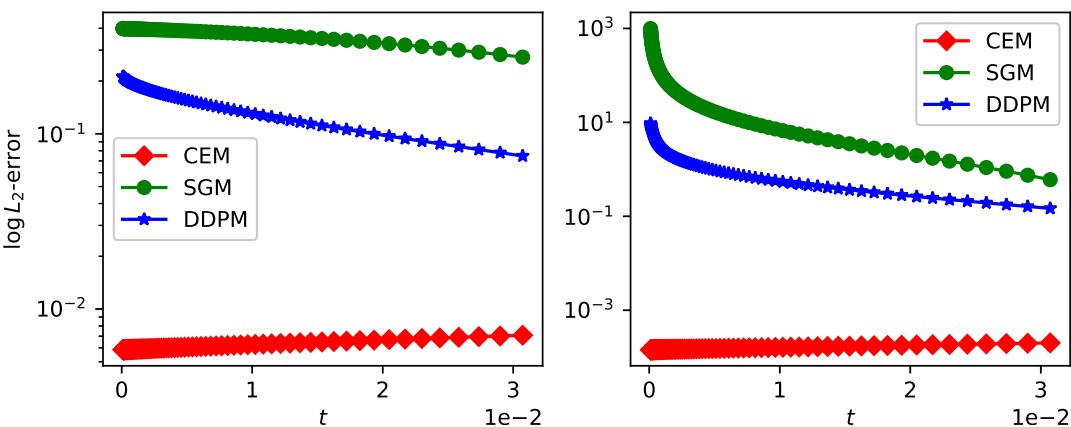

Figure 2: Error at a fixed point $X_{eva} = (1, -0.1)$. Red, proposed CEM: $e(f, f_\theta)$; Green, SGM: $e(S, S_\theta)$; Blue, DDPM: $e(\epsilon, \epsilon_\theta)$. (Left) first component of estimated function. (Right) second component of estimated function.

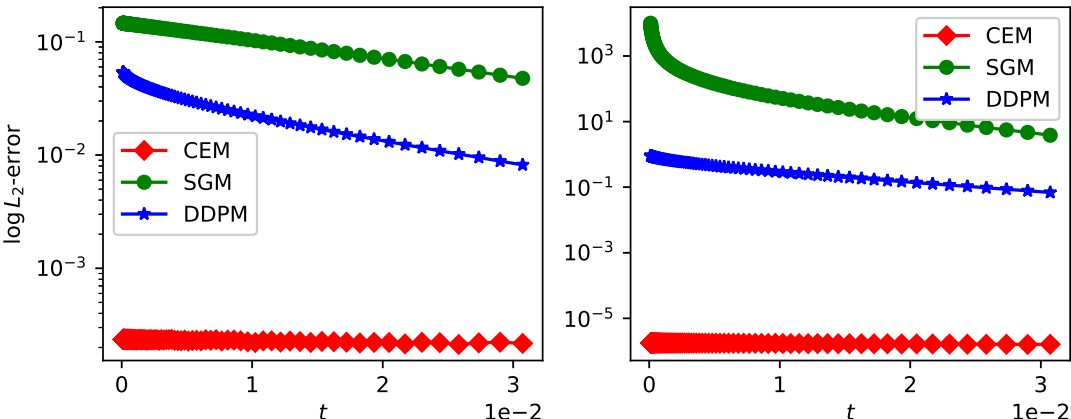

Figure 3: $L^2$-error with distribution $p$. Red, proposed CEM: $e_p(f, f_\theta)$; Green, SGM: $e_p(S, S_\theta)$; Blue, DDPM: $e_p(\epsilon, \epsilon_\theta)$. (Left) the first component of the model function. (Right) the second component of the model function.

**Curve distribution**  We now consider distributions with more complex geometries, and in particular a data distribution generated by $X = (U cos(U), U sin(U))$ where $U \sim \text{Unif}[1, 13]$. In Figure 4, we compare the distributions generated by the CEM, SGM, and DDPM. The singularities near $t = 0$ exhibited in Theorem 3.3 imply that errors only accumulate during the final few stages of the sampling process. The approximated stochastic dynamics primarily lead $X_t$ to a local neighborhood of the support of $p_{data}$, where most of the error is concentrated.

### 4.2    Replacing the network by analytical expressions

In a limited number of favorable settings, the diffusion coefficients $(v, D)$ that appear in the backward sampling process may be computed explicitly leading to an equally explicit expression for the conditional expectation (20). This bypasses the need to model $(v, D)$ by a neural network.

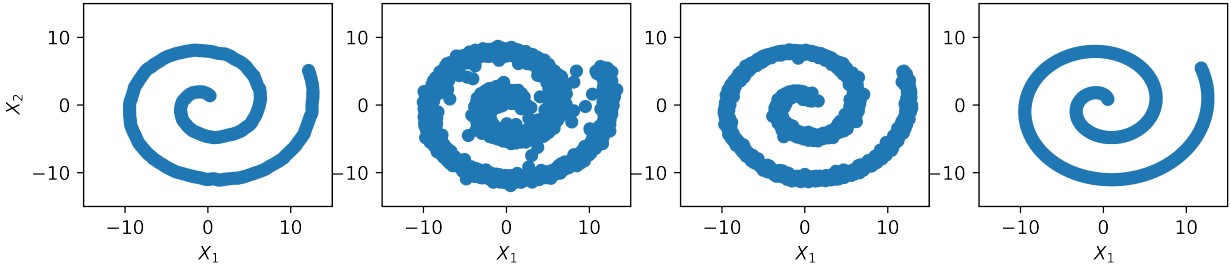

Figure 4: Curve distribution. From left to right: the proposed CEM, SGM, DDPM, and the ground truth. The network configuration is as follows: 3 hidden layers, each layer with 64 nodes, and **Tanh** as the activation function.

Table 1: Sampling five-point distribution in 2d space with analytic expression. Frequency of each point.

| POINTS | 1 | 2 | 3 | 4 | 5 |
|--------|--------|--------|--------|--------|--------|
| FREQ. | 0.2086 | 0.1924 | 0.2092 | 0.1977 | 0.1921 |

As an illustrative example, we generate 5 points (randomly) $X_{1:5}$ and set the target data distribution $p_{data} = \sum_{i=1}^{5} \delta_{X_i}$. We then obtain the following analytical expression derived in Appendix A.3,

$$E[X_0|X_t = X] = \frac{\sum_{i=1}^{5} X_0^{(i)} \exp\left(-\frac{\|X - X_0^{(i)} e^{-\frac{t}{2}}\|^2}{2(1 - e^{-t})}\right)}{\sum_{i=1}^{5} \exp\left(-\frac{\|X - X_0^{(i)} e^{-\frac{t}{2}}\|^2}{2(1 - e^{-t})}\right)}. \tag{75}$$

Figure 5 displays the backward process for 10000 samples generated by solving the backward SDE at the times $t = 10, 5.8718, 3.2356, 0.7518, 0.0216, 0$. Not surprisingly, the initial points sampled from a normal distribution are entirely "absorbed" into the target five-point distribution at the final sampling step $t = 0$. Table 1 counts the empirical frequencies (probability) of absorption by the five target points, which are very close to their theoretical value 0.2.

The interpretation is then twofold. (1) With an exact model of the target function in the training process and an exact solution of the SDE (18) in the sampling process, the resulting new samples accurately reproduce the original training data. This validates that the training process of diffusion model under the framework discussed in Section 3.1 is in fact a least square minimization which achieves optimal at conditional expectation equation 20. (2) When explicit expressions such as (75) are not available, this ideal accurate sampling of the training data can rarely be achieved in practice due to the imperfections in the neural network approximation. Only a simplified distribution is learned in practice, which enables the generalization abilities. See Figure 6 and the next section for the reconstructions obtained in the context of a distribution with four atoms for different neural nets trained to approximate $E[X_0|X_t = X]$.

### 4.3 Dependence on the model configuration

**Expressive power of network.** In the next example, we consider a point cloud distribution with four points $p_{data}(x) = \delta_{(1,-3)}(x) + \delta_{(1,-1)}(x) + \delta_{(1,1)}(x) + \delta_{(1,3)}(x)$. The analytical solution for $f$ is similar to that in (75). In Figure 6, we show the distribution generated from (24) after training for the CEM with different network configurations: a deep neural network (2 hidden layers, 16 nodes each layer) and a shallow neural network (1 hidden layer, 4 nodes each layer). This comparison illustrates that when the approximation power of the neural network is insufficient, then optimizing (65) only leads to a poorly approximated $f$ and hence an incorrect resulting sampling.

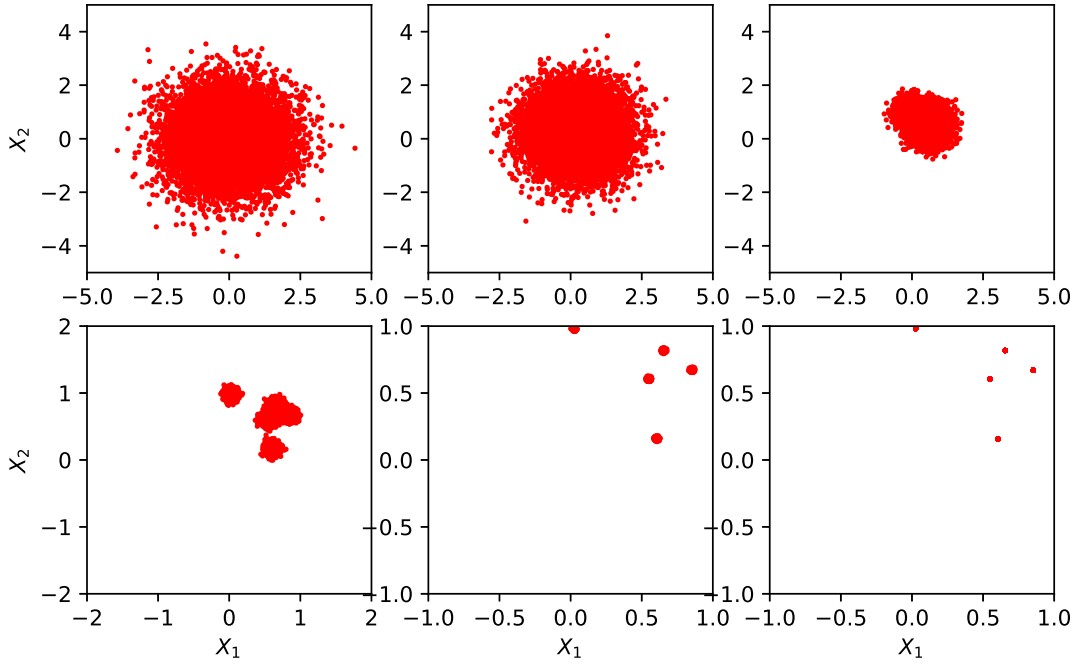

Figure 5: Generating Five-point distribution in 2d space by the analytic expression of the drift, scattering plot of sampling process for $t = 10, 5.8718, 3.2356, 0.7518, 0.0216, 0$.

We point out that for many practical distributions with bounded support, even though the proposed CEM target function $f$ is bounded uniformly in $t$, it may itself be extremely complicated. In practical applications of diffusion models, the new $f$ in CEM may have different structures compared to $\epsilon$ in DDPM and $S$ in SGM and this may require a design of the network that differs from the well-established U-Net in Ronneberger et al. (2015). Moreover, if the design of the network preserves the possible low dimensional structure of continuous data distribution, instead of the discrete samples, solving the backward process associated with the network modeled drift may *generalize* the distribution of the discrete sample to the continuous one. We leave this as future work.

**Training schedule** $t_1$   Yet another parameter to be determined in the training schedule proposed in Eq. (72) is $t_1$. In Figure 7, we consider a 20 points distribution in $\mathbb{R}^2$ and generate samples from Eq. (24) with analytical expression for various values of $t_1$. As a splitting scheme, (24) introduces numerical errors proportional to the time step. Since the final time step is $t_1$, we can see in Figure 7 that smaller $t_1$ results in lesser errors in the generated distribution. In practice, we do not recommend $t_1$ to be taken as too small as this introduces numerical instabilities when computing the final drift in (24).

**Aligning the training process by designing** $\lambda$   In order to further improve the effectiveness of training, it is also important to control the variance of the loss function at different times by judicious choices of $\lambda(t)$. In section 3.3, we propose to define $\lambda(t) = \frac{1}{e^t - 1}$ and the previous experiments have verified its validity. Recalling the loss function in CEM (65), we ensure that

$$\lambda(t)^{-1} \sim E_{X_0, X_t} \|X_0 - f(X_t, t)\|^2. \tag{76}$$

For a given explicit expression of $f$, we can numerically estimate the right-hand side at different times by sampling the forward process $X_t$. The estimation is denoted as $\lambda_{\text{true}}(t)$, as it reflects the potential small $t$ asymptotic regime of the variance in the right-hand side of (76).

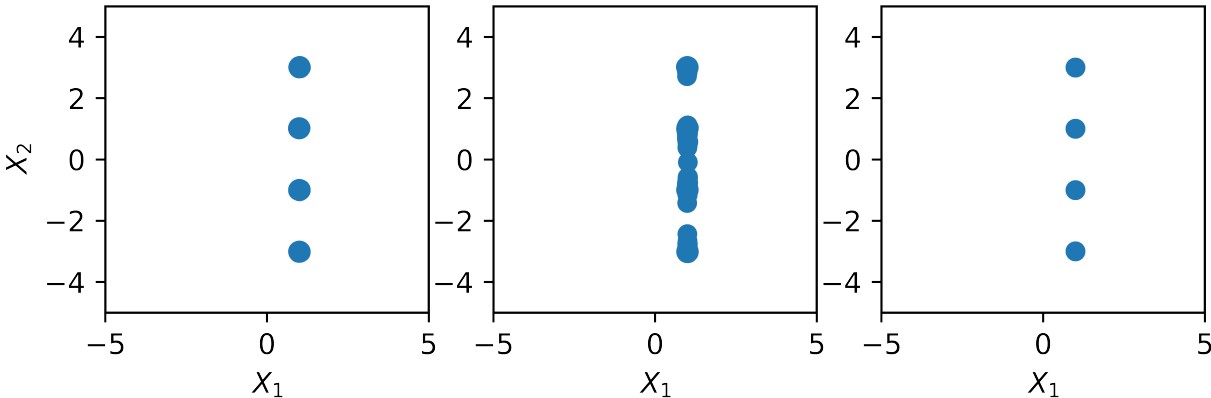

Figure 6: Four-point distribution. From left to right: Deep neural network, shallow neural network, and the ground truth. The Deep network configuration: 2 hidden layers, each layer with 16 nodes. The shallow network configuration: 1 hidden layer, each layer with 4 nodes. **Tanh** as the activation function of both.

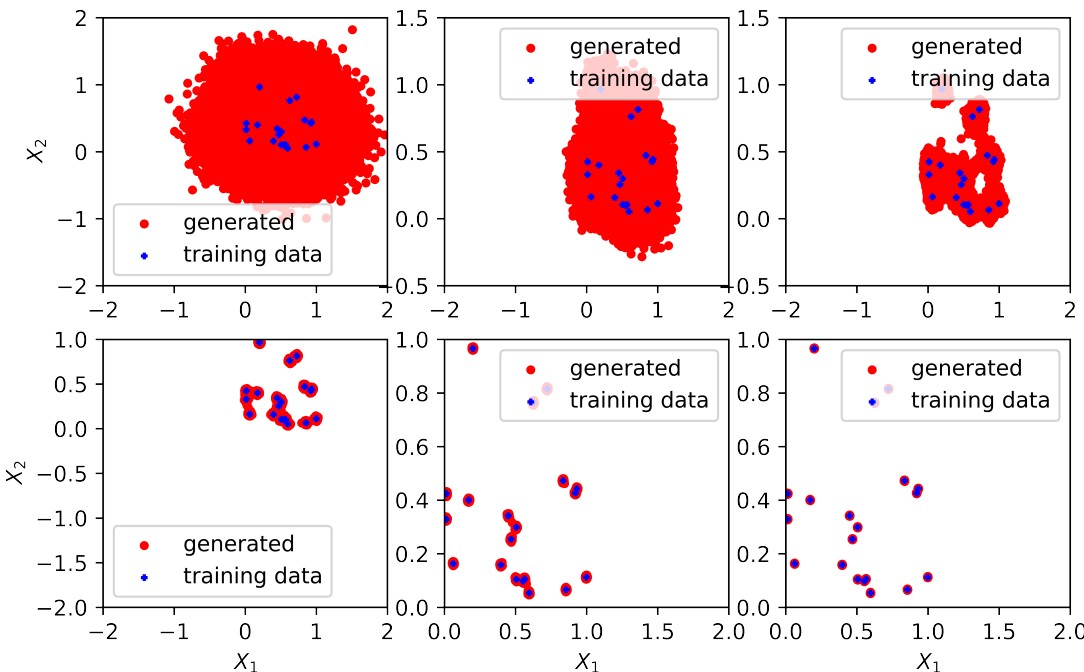

Figure 7: Generated samples with the analytic expression of drift for various $t_1$

In Figure 8, we revisit the case of Section 4.2 with a five-point target distribution $p_{data}$. The proposed $\lambda$ in Section 3.3 is $(e^t - 1)^{-1}$. We consider $\frac{1}{\lambda_{\text{guess}}(t)} = C(e^t - 1)$ with a free constant $C$ to fit the computed $\frac{1}{\lambda_{\text{true}}(t)}$. It can be seen on Figure 8 that $(e^t - 1)^{-1}$ captures the correct scale despite the minor perturbations introduced by the sampling. This result is another confirmation that our proposed method CEM may greatly improve training stability in some cases.

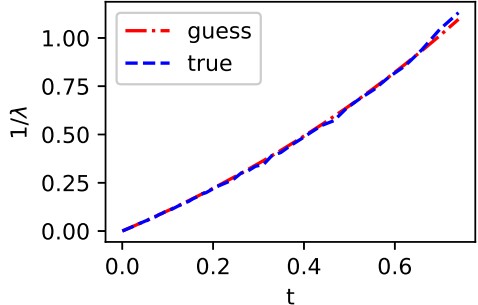

Figure 8: Fitting $\lambda^{-1}$. True: $E_{X_0,X_t}\|X_0 - f(X_t,t)\|^2$ by Monte Carlo; guess: $C(e^t - 1)$ for the best constant $C$.

## 4.4 Application to MNIST

In this subsection, we present the performance when applying our CEM to generate high dimensional distribution (MNIST, 9a). Comparing with previous examples, we replace the densely connected net by Unet Ronneberger et al. (2015) to model $\epsilon$ (66) of DDPM and $f$ of CEM separately. We apply **Adam** optimizer with a learning rate of 0.00002 and train each model with a batch size of 64 for 30 epochs. Both the forward process and the sampling process consist of 1000 steps, with a final time $T = 10$. Figure 9b and Figure 9c show that the generation of CEM and DDPM correspondingly. Limited by computing resources, this preliminary numerical result validates the potential sample generation capability of CEM for high dimensional distributions and shows the advantage of CEM over the original DDPM.

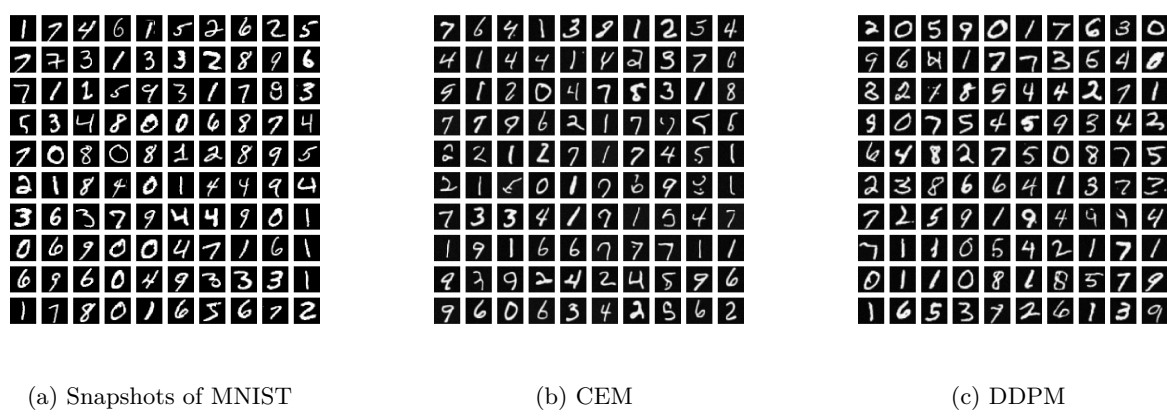

(a) Snapshots of MNIST          (b) CEM          (c) DDPM

Figure 9: Performance of CEM and DDPM on MNIST

## 4.5 Ablation Studies

**Impact of sampling schedule** Theorem 3.3 shows general existence of the singularities during the sampling process. An arbitrary sampling schedule may lead to numerical instabilities during solving reverse time SDEs. To this end, we take 20 time steps from $T = 10$ for the sampling process and compare the linear schedule, quadratic schedule and the proposed exponential schedule (72) in Figure 10. As expected, we can see that the exponential schedule significantly improves the sampling performance of CEM as a result of respecting the growth of scale of the drift. As an intermediate between linear and exponential, the quadratic schedule yields similar results to the exponential schedule, but with slightly inferior performance.

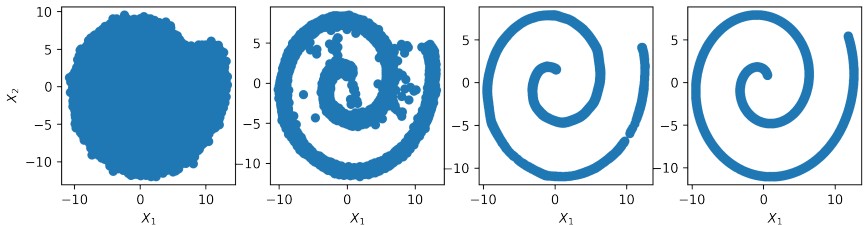

Figure 10: Comparison of different sampling schedules. From left to right: linear schedule, quadratic schedule, exponential schedule and ground truth. The network configuration is as follows: 3 hidden layers, each layer with 64 nodes, and **Tanh** as the activation function.

**Impact of weighting function** $\lambda$   The weighting function $\lambda$ in (65) is also a major impact factor for the performance and should be carefully designed for the training in order to normalize the training objective. We choose three different weighting functions $\lambda(t) = 1, \frac{1}{(e^t-1)^2}, \frac{1}{e^t-1}$ and compare their sampling performance in Figure 11. We can see that the proposed weighting function $\lambda(t) = \frac{1}{e^t-1}$ achieves a better sampling result than the other two functions.

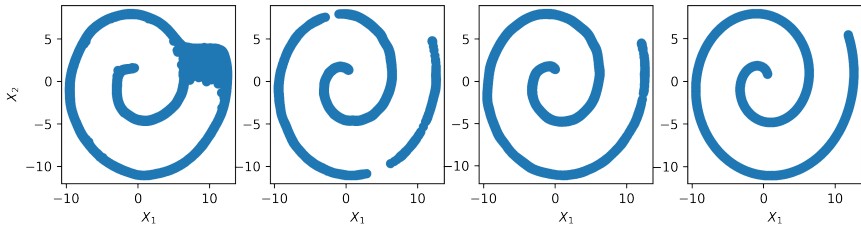

Figure 11: Comparison of different weighting functions. From left to right: constant weighting function, $\frac{1}{(e^t-1)^2}, \frac{1}{e^t-1}$ and ground truth. The network configuration is as follows: 3 hidden layers, each layer with 64 nodes, and **Tanh** as the activation function.

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

# A  Appendix

## A.1  Normal distribution

Consider a one-dimensional case. If $X_0$ is a normal distribution $\mathcal{N}(\mu, \sigma^2)$, the marginal density Eq. (19) becomes

$$p(t, X) = \frac{1}{Z\sqrt{2\pi}\sigma} \int \exp\left(-\frac{\|X - X_0 e^{-\frac{t}{2}}\|^2}{2(1 - e^{-t})}\right) \exp\left(-\frac{\|X_0 - \mu\|^2}{2\sigma^2}\right) dX_0$$

$$= \frac{1}{Z\sqrt{2\pi}\sigma} \int e^{-L(t, X, X_0)} dX_0. \tag{77}$$

Here the function $L(t, X, X_0)$ is denoted by:

$$
\begin{aligned}
L(t, X, X_0) &:= \frac{\|X - X_0 e^{-\frac{t}{2}}\|^2}{2(1 - e^{-t})} + \frac{\|X_0 - \mu\|^2}{2\sigma^2} \\
&= \frac{\sigma^2 \|X - X_0 e^{-\frac{t}{2}}\|^2 + \|X_0 - \mu\|^2 (1 - e^{-t})}{2\sigma^2 (1 - e^{-t})} \\
&= \frac{A(t) \|X_0\|^2 - B(t, X) X_0 + \sigma^2 \|X\|^2 + \mu^2 (1 - e^{-t})}{2\sigma^2 (1 - e^{-t})} \\
&= \frac{\sigma^2 \|X\|^2 + \mu^2 (1 - e^{-t})}{2\sigma^2 (1 - e^{-t})} + \frac{A\|X_0 - \frac{B}{2A}\|^2 - \frac{B^2}{4A}}{2\sigma^2 (1 - e^{-t})} \\
&= \frac{\sigma^2 \|X\|^2 + \mu^2 (1 - e^{-t})}{2\sigma^2 (1 - e^{-t})} - \frac{B^2}{8A\sigma^2 (1 - e^{-t})} + \frac{A\|X_0 - \frac{B}{2A}\|^2}{2\sigma^2 (1 - e^{-t})},
\end{aligned} \tag{78}
$$

where the function $A(t) = \sigma^2 e^{-t} + 1 - e^{-t}$ and $B(t, X) = 2X\sigma^2 e^{-\frac{t}{2}} + 2\mu(1 - e^{-t})$. Therefore, we can rewrite the marginal density (77) as

$$
\begin{aligned}
p(t, X) &= \frac{1}{Z\sqrt{2\pi}\sigma} \exp\left(\frac{B^2}{8A\sigma^2 (1 - e^{-t})} - \frac{\sigma^2 \|X\|^2 + \mu^2 (1 - e^{-t})}{2\sigma^2 (1 - e^{-t})}\right) \int \exp\left(-\frac{A\|X_0 - \frac{B}{2A}\|^2}{2\sigma^2 (1 - e^{-t})}\right) dX_0 \\
&= \frac{1}{\sqrt{2\pi A}} \exp\left(\frac{B^2}{8A\sigma^2 (1 - e^{-t})} - \frac{\sigma^2 \|X\|^2 + \mu^2 (1 - e^{-t})}{2\sigma^2 (1 - e^{-t})}\right) \frac{\sqrt{A}}{Z\sigma} \int \exp\left(-\frac{A\|X_0 - \frac{B}{2A}\|^2}{2\sigma^2 (1 - e^{-t})}\right) dX_0 \\
&= \frac{1}{\sqrt{2\pi A}} \exp\left(\frac{B^2}{8A\sigma^2 (1 - e^{-t})} - \frac{\sigma^2 \|X\|^2 + \mu^2 (1 - e^{-t})}{2\sigma^2 (1 - e^{-t})}\right).
\end{aligned} \tag{79}
$$

Subsequently, we have

$$\log p(t, X) = \log\left(\frac{1}{\sqrt{2\pi A}}\right) + \frac{B^2}{8A\sigma^2 (1 - e^{-t})} - \frac{\sigma^2 \|X\|^2 + \mu^2 (1 - e^{-t})}{2\sigma^2 (1 - e^{-t})}. \tag{80}$$

Substituting $A(t) = \sigma^2 e^{-t} + 1 - e^{-t}$ and $B(t, X) = 2X\sigma^2 e^{-\frac{t}{2}} + 2\mu(1 - e^{-t})$ into (80) yields

$$
\begin{aligned}
\nabla_X \log p(t, X) &= \frac{B\nabla_X B}{4A\sigma^2 (1 - e^{-t})} - \frac{X}{1 - e^{-t}} \\
&= \frac{4\sigma^4 e^{-t} X + 4\mu\sigma^2 e^{-\frac{t}{2}} (1 - e^{-t})}{4\sigma^2 (1 - e^{-t})(\sigma^2 e^{-t} + 1 - e^{-t})} - \frac{X}{1 - e^{-t}} \\
&= \left[\frac{\sigma^2 e^{-t}}{(1 - e^{-t})(\sigma^2 e^{-t} + 1 - e^{-t})} - \frac{1}{1 - e^{-t}}\right] X + \frac{\mu e^{-\frac{t}{2}}}{\sigma^2 e^{-t} + 1 - e^{-t}} \\
&= \frac{-X + \mu e^{-\frac{t}{2}}}{\sigma^2 e^{-t} + (1 - e^{-t})}.
\end{aligned} \tag{81}
$$

Thus, the function $\nabla_X \log p(t, X)$ is not singular at $t = 0$. This agrees with $\nabla_X \log p(t, X) = -X$ when $\mu = 0$ and $\sigma = 1$.

## A.2 Distribution supported on a low dimensional manifold

If $X_0$ is a normal distribution $\mathcal{N}(0,1)$ and $Y_0$ is a $\delta_0$-distribution, the marginal density in (19) becomes

$$
\begin{aligned}
p(t,X,Y) &= \frac{1}{Z} \int \int \exp -\frac{(X - X_0 e^{-\frac{t}{2}})^2 + (Y - Y_0 e^{-\frac{t}{2}})^2}{2(1 - e^{-t})} \rho(X_0, Y_0) dX_0 dY_0 \\
&= \frac{1}{Z} \int \int \exp -\frac{(X - X_0 e^{-\frac{t}{2}})^2 + (Y - Y_0 e^{-\frac{t}{2}})^2}{2(1 - e^{-t})} \rho(X_0) \delta_0(Y_0) dX_0 dY_0 \\
&= \frac{1}{Z\sqrt{2\pi}} \int \exp -\frac{(X - X_0 e^{-\frac{t}{2}})^2 + Y^2}{2(1 - e^{-t})} e^{-\frac{X_0^2}{2}} dX_0 \\
&= \frac{1}{Z\sqrt{2\pi}} \int \exp -\frac{(X_0 - X e^{-\frac{t}{2}})^2 + X^2 + Y^2 - X^2 e^{-t}}{2(1 - e^{-t})} dX_0 \\
&= \frac{\sqrt{1 - e^{-t}}}{Z} \exp \frac{X^2 e^{-t} - X^2 - Y^2}{2(1 - e^{-t})}.
\end{aligned}
\tag{82}
$$

Therefore,

$$
\log p(t,X,Y) = \log(\frac{\sqrt{1 - e^{-t}}}{Z}) + \frac{X^2 e^{-t} - X^2 - Y^2}{2(1 - e^{-t})},
\tag{83}
$$

and

$$
\nabla_X \log p(t,X,Y) = -X,
\tag{84}
$$

$$
\nabla_Y \log p(t,X,Y) = -\frac{Y}{1 - e^{-t}}.
\tag{85}
$$

## A.3 Point cloud distribution

We now derive the analytic expression for $E[X_0|X_t = X]$ when $X_0$ is drawn from a point cloud. Suppose that the number of points is $K$ (denoted by $\{X_0^{(i)}\}_{i=1}^K$), then

$$
E[X_0|X_t = X] = \frac{\frac{1}{Z} \int X_0 \exp\left(-\frac{\|X - X_0 e^{-t/2}\|^2}{2(1 - e^{-t})}\right) p_{data}(X_0) dX_0}{\frac{1}{Z} \int \exp\left(-\frac{\|X - X_0 e^{-t/2}\|^2}{2(1 - e^{-t})}\right) p_{data}(X_0) dX_0} = \frac{\sum_{i=1}^K X_0^{(i)} \exp\left(-\frac{\|X - X_0^{(i)} e^{-\frac{t}{2}}\|^2}{2(1 - e^{-t})}\right)}{\sum_{i=1}^K \exp\left(-\frac{\|X - X_0^{(i)} e^{-\frac{t}{2}}\|^2}{2(1 - e^{-t})}\right)},
\tag{86}
$$

where $Z$ is a normalizing factor that depends on $t$ and the dimension of $X_0$.

