# OpenReview forum: "Mathematical analysis of singularities in the diffusion model under the submanifold assumption"
_TMLR — Rejected by TMLR_

### Review · Reviewer_fqTD · 2023-05-16

**Summary Of Contributions:**

The paper presents mathematical observations about score-based generative models (SGMs). The authors notably observe that the score function experiences asymptotic blow-up at the initial time for data distributions supported on lower-dimensional submanifolds. This issue presents challenges for learning the score function and sampling from the reverse SDE. To mitigate this issue, the authors propose a modified training objective.

**Audience:**

No

**Broader Impact Concerns:**

None.

**Claims And Evidence:**

Yes

**Requested Changes:**

Conduct a far more thorough literature search. The authors need to consider the novelty of their contributions more carefully. I understand that the acceptance criteria of TMLR does not require substantial originality over prior works, but in this case the findings of these paper have directly appeared in prior works which is problematic.

Top of pg. 2: The Langevin diffusion should correspond to $(v, D) = (-\nabla V, I)$.

Eq. (7): It should possibly be $\widetilde{X_k}$, not $\widetilde{X_{k-1}}$.

Top of pg. 4: Correct the grammar in "when for T large".

Bottom of pg. 4: Correct the typo in "such denoiser is only though experiment".

Section 3.1: This section is already well-covered in the literature, e.g., section B.2 of [V. De Bortoli, Convergence of denoising diffusion models under the manifold hypothesis.]

Eq. (17): This equation does not represent the analytic solution of the Langevin equation, it only captures the marginal law.

The comment underneath eq. (17) about the advantage of expressing SGMs as an SDE is also well-known. For example, the works [S. Chen, S. Chewi, J. Li, Y. Li, A. Salim, A. R. Zhang, Sampling is as easy as learning the score: theory for diffusion models with minimal data assumptions; H. Chen, H. Lee, J. Lu, Improved analysis of score-based generative modeling: user-friendly bounds under minimal smoothness assumptions; H. Lee, J. Lu, Y. Tan, Convergence of score-based generative modeling for general data distributions] have provided convergence guarantees for SGMs based on SDE analysis.

Expressing the score function as a conditional expectation is also very well-known and should not be claimed as a contribution of this paper. Training objectives such as (65) have also appeared previously in the literature. See, e.g., the discussion in S. Chen et al., although these observations are older.

Assumption (H2) does not make sense; what is meant by $B_\varepsilon \subseteq B_{\varepsilon_0}$?

Theorem 3.3 does not seem particularly novel. The works of De Bortoli and Lee, Lu, Tan mentioned above all mention the blow-up as t tends to zero and in fact explicitly incorporate this into their analysis.

Remark 3.5: The uniqueness assumption is an assumption about the manifold on which the data distribution is supported and there is no sense in which this assumption holds "almost surely". The uniqueness assumption is also quite restrictive, ruling out many important examples of manifolds.

Theorem 3.6: I assume that "the score function remains bounded" means simply that for any fixed X, the score function remains finite as t tends to 0. This is a trivial statement and does not require $\rho$ to have the particular form of a Gaussian convolution. Instead, any distribution $\rho$ which is smooth enough such that $\nabla \log \rho$ exists should avoid the singularity, because the score function converges to $\nabla \log \rho$ as $t\searrow 0$.

**Strengths And Weaknesses:**

Strengths: The proposed modified training objective mitigates the singularity issue and is supported by numerical experiments.

Weaknesses: The major weakness is that the paper's contributions overlap significantly with existing literature; pretty much all of the observations made in this paper are already **well-known**. Additionally, some parts of the paper contain errors and unclear statements. Given these issues, the paper requires substantial revisions. Therefore, I strongly recommend rejection in its current form.

---

> ### Author Response · Authors · 2023-06-15
>
> Weaknesses: We thank the reviewer for his critical suggestions and for pointing out some typos in our paper. But we can only partially agree with the reviewer's comments. Especially since we do not agree with the novelty concerns. First, our representation of conditional expectation reveals the goal of the training process is equivalent to finding the network representation of $\nabla \log \rho$. This is due to the conditional expectation is the global minimizer of $L_2$ misfit functional, Eq. (23). In this regard, our follow-up asymptotic (blow-up) analysis applies generally to diffusion models that have a continuous SDE representation under the framework in Section 3.1. Second, we would like to point out all the reference that provided by the reviewer only reveals the upper bound of possible blow-up behavior or take the blow-up as an assumption. In contrast, our theory provides mathematically rigorous analyses of asymptotic behaviors, which include sharp upper bound and lower bound. In particular, our theory shows the existence of blow-ups under the general sub-manifold assumption.  This is one of the major contributions of our work.
>
> We respond to the reviewer's concerns one by one as follows.
> 1. Thanks to the reviewer for pointing out this error. We will correct it.
> 2. Eq. (7) is the backward process. Therefore, the variable $\tilde{X}_{k-1}$ on the right-hand side represents the following step in the sampling iteration.
> 3. Thanks to the reviewer for pointing out this grammar error. We will correct it.
> 4. Thanks to the reviewer for pointing out this typo. We will correct it.
> 5. We agree that the SDE framework is unified in prior works. This time framework includes not only discrete and continuous time but also the time transformation relationships between different SDE in diffusion models (says, the drift and diffusivity of the SDE take the forms $g(t)x$ and $\sqrt{g(t)}$, respectively). While in Section 3.1, we aim at providing such a unified framework as a fundamental basis for presenting the asymptotic analysis in the following sections. In this regard, we can point out that the singularities exist generally in diffusion models following such a framework. We have revised the first paragraph of Section 3.1 accordingly. Refer to the response regarding weaknesses as well.
> 6. We have modified Eq. (17) accordingly.
> 7. We agree with the reviewer's comment. These advantages are well known, but a complete description is necessary for our following asymptotic analysis.
> 8.  The conditional expectation in Eq. (20) is not a new result. While by realizing the conditional expectation is the minimizer of $L_2$ misfit functional, we show forward training process is modeling (approximating) Eq. (20) by the network (score function). Also, it is the key to realizing the singularity of the score function, and our work is based on this key insight.  We decide to remove the first item in the statement, `Main contributions of this work'. In addition, we carefully read the paper [S. Chen et al.,], but we cannot find the training objective (65) in their paper. Training objective (65) is derived from our theory. Another paper proposed a similar training objective [T. Karras, M. Aittala, T. Aila, and S. Laine. Elucidating the design space of diffusion-based generative models], but they cannot theoretically explain why this training objective is better than the classical one.
> 9. We have modified (H2) as `Let $B_{\varepsilon}=\{y\in \Omega: \|y-X\|<\|y_X-X\| + \varepsilon\}$, which is decreasing set series as $\varepsilon\to 0$. We assume there exists $0<\varepsilon_0\ll1$, such that for $y\in B_{\varepsilon_0}$, there exists a local coordinate chart, $z\to y(z)\in B_{\varepsilon_0}\subset\Omega$, ...'.
> 10. Thanks to the reviewer for bringing this paper to our attention. After reading this paper carefully, we would like to point out a couple of differences below. We agree that the works of De Bortoli and Lee, Lu, Tan mentioned above all mention the blow-up as t tends to zero. With noticing blow-up behavior experimentally, their derivation is based on the assumed existence of blow-up as t tends to zero, see Assumption A3. Especially, their theory does not prove such existence. In contrast, our theory provides mathematically rigorous analyses of asymptotic behavior at the final time, which includes both sharp upper and lower bound. The upper bound is related to A3.   This is one of the major contributions in our work.
> 11. The word `almost surely' is defined in measure theory, which means the measure of the n-dimensional manifold $\Omega$ is equal to zero in the d-dimensional space ($n<d$). Therefore, the uniqueness assumption holds for arbitrary $X\in\mathbb{R}^d\backslash\Omega$.  In addition, we would like to point out real world data (human genes, climate patterns, and images as discussed in Remark 3.5 in the manuscript) are in general lower dimensional as they have to follow some symmetries as physic laws.

---

> ### Author Response · Authors · 2023-06-15
>
> 12. We agree that  if $\nabla\log \rho$ (limit as $t\to 0$) exists, the singularities are surpassed. While Theorem 3.6 provide a sufficient condition for such a limit to exist. The necessary and sufficient condition for this problem is not trivial. From our analysis of Theorem 3.6, it, in fact, turns out to be the necessary and sufficient condition for $\rho_0$ to be a positive density function. Note that $\rho_0$ can be attained by Fourier transform of $\hat{\rho_0}:=\hat{\rho}/\hat{\mu_1}$. While the necessary and sufficient condition for `the Fourier transform of any function in $R^d$ is positive' is unknown in math.

---

> > ### Comment · Reviewer_fqTD · 2023-08-05
> > **Response**
> >
> > Thank you for your response. I would like to point out that Reviewer 5Q8o also left a detailed review pointing out many of the same novelty considerations that I raised. I fully agree with that reviewer's concerns and I would appreciate seeing your response to those points.

---

> > > ### Author Response · Authors · 2023-08-11
> > >
> > > Thank you for your comments. We have replied to Reviewer 5Q8o and clarified the novelty of our work in the revised paper. Especially, we have included a brief discussion of relate work at the end of Section 1. Please see our revised paper.

---

> ### Author Response · Authors · 2023-09-18
>
> Thanks for fqTD's modified feedback, while we are uncertain about what the specific changes has been made to the review on 14 Sept. Meanwhile, we have carefully reviewed the current review and our responses. Beside that we would like the reviewer to consider the theoretical contribution of the manuscript, namely the rigorous proof of existence of singularity in general diffusion model, we have provided itemised responses to each of fqTD's comments. If you have any further questions or requirement, please feel free to let us know.

---

### Review · Reviewer_5Q8o · 2023-06-18

**Summary Of Contributions:**

This paper aims at providing some theoretical insights in diffusion models. In particular, the authors first claim to unify Denoising Diffusion Probabilistic Models (DDPMs) and Score-Based Generative Models (SGMs). Then, they move onto showing that the score has a singularity if the data distribution is compactly supported. To deal with that issue they provide a parameterization of the score function. Finally, their findings are illustrated with low dimensional experiments.

**Audience:**

Yes

**Claims And Evidence:**

No

**Requested Changes:**

The authors should address the list of weaknesses. In particular:

* The authors should better position their work w.r.t. the existing literature (link between SGM/DDPM, singularity of the score, parameterisations).

* Provide more in depth experiments. Higher dimensional experiments and ablation studies should be conducted.

**Strengths And Weaknesses:**

Strengths:

* Provide some mathematical insights on diffusion models.

Weaknesses:

* The first claim of the authors is that "By exploring a unified framework based on reverse-time SDE, we show that the training process of DDPM and SGM is approximating a conditional expectation functional (score function)". This link was already established in works like the ones of [1]. In particular, Appendix B of [1] is specifically dedicated to drawing a bridge between DDPM and SGM "Below we provide detailed derivations to show that the noise perturbations of SMLD and DDPM are discretizations of the Variance Exploding (VE) and Variance Preserving (VP) SDE". The authors should discuss this work and how their derivations are different in details. What are the specific insights provided by the authors and how can they be leveraged?

* The claim "Despite its success, the sampling process for diffusion models is extremely slow and the computational cost is high." is way too strong. This ignores a large span of research in diffusion models which is dedicated to finding faster schemes for the sampling and/or training of diffusion models. In particular, distillation models such as [2,3,4] or the better discretization models of [5,6] (to cite a few) have considerably enhanced the speed of diffusion models. This is not discussed in the paper.

* The observation that the score is singular in the presence of compactly supported data is not new and is first found in [10,7] which provide guarantees for the diffusion model to be supported on the data manifold. Similarly [8] studied the convergence of diffusion models precisely under the manifold hypothesis and derived quantitative convergence bounds. Finally, [9] studies the parameterisation and the approximation properties of the score under the manifold hypothesis. None of these theoretical works are discussed by the authors. It is not clear to me that the present analysis brings anymore insight.

* There is a huge literature on the parameterisation of the score in diffusion model. The "new loss function" defined in (65) is not new and can be found in [3]. The $x_0$ parameterisation is also discussed in [11], see Equation (15). In [3] there is a discussion on the use of different $\lambda(t)$ (this is in the context of $v$-prediction). The loss on the reconstruction instead of the score has been routinely used in other works such as [12] (in the context of discrete models).

* Experiments are only low dimensional. I am quite suspicious of Figure 1 for instance as a better fine tuning of SGM and DDPM could lead to better results. Code is not available to reproduce the results. There are several hyperparameters for each of these methods and a proper ablation study should be conducted.

* The experiment conducted in Section 4.2 can also be found in [13] on CIFAR10. "This numerically supports the use of the conditional expectation to explain the model S". I don't understand this claim. The link between the score and conditional expectation is well-known. For a variational point of view on this, see [14].

* There is no conclusion or related work.

[1] Song et al. (2020) -- Score-Based Generative Modeling through Stochastic Differential Equations

[2] Meng et al. (2022) -- On Distillation of Guided Diffusion Models

[3] Salimans and Ho (2022) -- Progressive Distillation for Fast Sampling of Diffusion Models

[4] Luhman and Luhman (2021) -- Knowledge Distillation in Iterative Generative Models for Improved Sampling Speed

[5] Jolicoeur-Martineau et al. (2021) -- Gotta Go Fast When Generating Data with Score-Based Models

[6] Zhang and Chen (2022) -- gDDIM: Generalized denoising diffusion implicit models

[7] Pidstrigach (2022) -- Score-Based Generative Models Detect Manifolds

[8] De Bortoli (2022) -- Convergence of denoising diffusion models under the manifold hypothesis

[9] Chen et al. (2023) -- Score Approximation, Estimation and Distribution Recovery of Diffusion Models on Low-Dimensional Data

[10] De Bortoli et al. (2021) -- Diffusion Schrödinger Bridge with Applications to Score-Based Generative Modeling

[11] Ho et al. (2020) -- Denoising diffusion probabilistic models

[12] Campbell et al. (2022) -- A Continuous Time Framework for Discrete Denoising Models

[13] Peluchetti (2022) -- Non-Denoising Forward-Time Diffusions

[14] Huang et al. (2021) -- A Variational Perspective on Diffusion-Based Generative Models and Score Matching

---

> ### Author Response · Authors · 2023-08-11
>
> We thank the reviewer for his critical suggestions. We respond to the reviewer's concerns one by one as follows. We have posted a revised manuscript.
>
> 1. We agree that the SDE framework is unified in prior works. This time framework includes not only discrete and continuous time but also the time transformation relationships between different SDE in diffusion models (says, the drift and diffusivity of the SDE take the forms $g(t)x$ and $\sqrt{g(t)}$, respectively). While in Section 3.1, we aim at providing such a unified framework as a fundamental basis for presenting the asymptotic analysis in the following sections. In this regard, we can rigorously show that the singularities exist generally in diffusion models following such a framework. We have revised the main contribution and the first two paragraphs of Section 3 accordingly.
>
> 2. We thank the reviewer for pointing out these related references. We have included some of them in our paper. But, more importantly, our results highlight the need for careful design of methods to enhance the speed of diffusion models, as the singularity of the score function occurs near $t=0$ during the sampling process (for singular data distributions). That is a contribution in our paper.
>
> 3. We agree the reviewer's comments. The singularity of the score function is already discussed in prior works and we {have included} the references that point out by the reviewer. However, we would like to point out a couple of differences below (we also include a brief discussion of related work at the end of Section 1). We agree that the works of De Bortoli [8, 10] mentioned above all mention the blow-up as $t$ tends to zero. While with noticing blow-up behavior experimentally, their derivation is based on the assumed existence of blow-up as t tends to zero, see Assumption A3 in [8]. Especially, their theory does not prove such existence. In contrast, our theory provides mathematically rigorous analyses of asymptotic behavior at the final time, which includes both sharp upper and lower bound. The upper bound is related to Assumption A3 in [8]. More precisely, we found that the singular behavior of the score function is $\frac{1}{t}$, which is a sharper result than [7] (they proved that $\|\nabla\log(p_t)\|\gtrsim\frac{1}{\sqrt{t}}$ for the Brownian motion example). Besides, [9] should be considered as a parallel work by another group and they require a 'linear assumption', which is a special case of our theory, on the dataset (the data point x satisfies $x=Az$, where $z$ is referred to as the latent variable. Please see Section 1 in [9]. )
>
> 4. We thank the reviewer for pointing out these related references. We have discussed them in our updated manuscript. While we would like to point out that they used these strategies based on experiments, not a rigorous mathematical analysis at least. Unlike them, we provide a theoretical foundation to support these strategies.
>
> 5. A better fine-tuning of SGM and DDPM may lead to better results. While we compare these results in a fixed setting. We did not conduct fine-tuning for any model. Moreover, our CEM, which respects the singularities, does not require better fine-tuning to obtain a good result than SGM and DDPM. The code will be shared on GitHub after the anonymous review. Ablation studies for the sampling schedule and the weighting function are conducted in the revised paper.
>
> 6. The result of the experiment in Section 4.2 shows that with an exact analytical expression of conditional expectation (20) substituting into the sampling process the diffusion model does not have the ability to generate diverse samples. This verifies the link between score and conditional expectation. Our example serves this goal as the explicit expression (75) for (20) is derived in Appendix A.3 and in the experiment, after the substitution, the sampling process indeed generates a uniform distribution over five training data points. Our example is more intuitive compared to one in  [13]. Combining this validation result in Section 4.2 with the blow-up analysis in the previous section, we can see the privilege of CEM that bypass the singularities during training for more complex models.
>
> 7. We have summarized the discussions mentioned above to related work in our revised paper.
>
> Overall, we have added a related work part to discuss the difference between our work and the existing literature (the main points have been covered in the above reply). Moreover, we have conducted ablation studies for the sampling schedule and the weighting function and verified the ability of CEM on the MNIST dataset.

---

### Review · Reviewer_NEQv · 2023-07-31

**Summary Of Contributions:**

The paper under review studies the singularity of the score/loss function for the training process of the diffusion models. The analysis is based on the assumption that the observation of $X_t$ states in a sub-manifold of the whole space. Based on the main assumptions (H1) and (H2), the asymptotic expansion of the score function is derived, which shows the score function blows up as $t\rightarrow 0$. Based on such an observation, an alternative loss (CEM) is constructed, which outperforms the SGM and DDPM models in several experiments.

**Audience:**

Yes

**Claims And Evidence:**

Yes

**Requested Changes:**

Here are some minor comments:

1. For general forward SDE, could authors make any comments if any tools or results could be obtained? Does the new model/loss could be applied as well?

2. Page 10, Let ..be.

3. For distribution supported on a low dimensional manifold, how does this affect the drift in the diffusion process/SDE (both continuous and discrete case)? If the gradient is only a sub-gradient in the drift term, this is indeed related to hypo-elliptic SDEs, and the convergence of such type SDEs are not trivial at all, does any of the results in this paper depends on/connects to hypo-elliptic SDE? In this case, how do we know about the convergence rate?



**Strengths And Weaknesses:**

The main technique is based on analytical solution of the law of OU process, and explicit computation of the conditional expectation are derived. The proof of Theorem 3.3 is based on explicit computation and asymptotic analysis. Which is generally related to Laplace method and is interesting on its own. The new model based on condition expectation indeed provide a solution when there is a singularity when time $t\rightarrow 0$. Although the forward process is a simple OU process, which does not cover all forward SDEs, the explicit computation provides a nice and clear presentation of the potential singularity. The example of 1d line gaussian and 2d point cloud distribution are good examples for such CEM model.

Overall, the paper is well written and the idea is presented clearly.

The limitation of the paper is the explicit expression of the law of OU process. If one use the general Laplace method, to which expense one could recover the results in this paper in a more general setup. In particular, as the sub-manifold assumption, how do we know about the smoothness of the density along time $t$.

How much effort are required if one want to conduct experiments in a high dimension example, regarding the explicit computation similar to  the low dimensional experiment.

For a more general forward SDE, it there a method/way to obtain the gradient estimate of the function $\log p(t,X)$.

---

> ### Author Response · Authors · 2023-08-11
>
> We would like to thank the reviewer for that comment.
>
> 1.  As pointed out by remark 3.4, the proof provided in the manuscript can be considered as a generalized Laplace method. While such a setup would require a global coordinate chart representation of the density, otherwise the integral in (55) is not rigorously defined. In contrast, our proof is explicit and only requires a local coordinate chart at $y_X$. This is more useful in cases where data are clustered while each cluster has its own local dimension.
>
> 2. We have conducted an experiment on the MNIST dataset to verify the performance of CEM. Roughly speaking, we use almost the same setup as the low-dimensional experiment, except that we replace the fully connected neural network with Unet to approximate the score function. Details have been presented in the revised paper.
>
> 3. This is an interesting question. Our current analysis bases on OU process as the forward SDE. In this way, the density $p$ has {an} explicit expression, see (20). For a more general forward SDE, if still we have the explicit expression for $p$, our estimate method can then be generalized. We will leave this kind of analysis as future work.
>
> 4. For a more general forward SDE, if still we have the explicit expression for $p$, our estimate method can then be generalized. In the analyses of the manuscript, we show the score function exhibits singularities in case of distribution is lower dimensional and the forward process is OU. Such analyses framework shall apply generally if the analytic solutions of forward SDE and distributions $p$ is given. In future, we will seek for other types of forward SDE that could bypass the singularities during sampling.
>
> 5. Thanks for pointing out this typo, we have corrected it.
>
> 6. We agree with the reviewer's observation. Indeed, our manuscript aims at pointing out that the singularity is unavoidable given the low dimensional supported distribution under the current diffusion model framework (more precisely, the framework unified in Section 3.1). In this way, the convergence of SDE and the existence of a convergent SDE integrator are not trivial. Moreover, in the real-world application, e.g. generating pictures, the local representation of the data manifold is unknown and hence there is no automatic way to set up degenerated diffusion coefficient matrix. As one of the future directions, we are considering some structure-preserving methods to resolve the issue, which is not discussed in the current submission.

---

### Decision · Action_Editor_K4vw · 2023-10-28

**Recommendation:** Reject

**Comment:**

The paper received mixed reviews, with two out of three reviewers noting that most of the paper's contributions can be found in the existing literature on diffusion models. The third reviewer also highlighted the lack of perspective on where the paper stands in the current literature.

I share the same concerns as the reviewers. Currently, the paper lacks rigor in the conditions it considers and its establishment of the main result. It is not clear what novelty it brings compared to existing results on the explosion of the score function.

In addition, the proposed methodology appears not to be new, and the experimental illustrations have significant weaknesses.

For these reasons, I have decided to reject the paper.

I would suggest that the authors consider the following improvements:
- Emphasize their contributions in relation to existing results.
- Improve the formalism and rigor of condition A2, as well as the proof of their main result (Theorem 3.3).
- Provide better clarification of their methodological contributions and improve their experiments.

**Audience:**

Since diffusion models are now very popular in generative modelling, the problem addressed in this paper is relevant to TMLR audience.
However, in my opinion, the paper (in its current form at least), holds limited interest for the community, partly due to its poor writing (see my comments below).

**Claims And Evidence:**

Diffusion generative models consist in learning a family of score functions $(\nabla \log p_t  :  t \in [0,T])$ associated to a noising process starting from the data distribution. It has been observed that the learning procedure is as difficult as $t$ is close to $0$. The paper under review aim to understand this phenomenon and analyzes the behavior of the score $(\nabla \log p_t :  t \in [0,T])$ as $t$ approaches $0$. Under the assumptions that the data distribution lives on a sub-manifolds of $\mathbb{R}^d$ with dimension $< d$, and otherwise technical conditions, the authors  roughly shows that $||\nabla \log p_t(x)||$ goes to $\infty$ as $t \to 0$ and more precisely derives an asymptotic expansion of the score function.  In light of this observation, the paper introduces an alternative loss function (CEM), which surpasses the performance of the SGM and DDPM models in various experiments.

While this point has not been raised by the reviewers, I found the writing of the technical assumption A2 and its use very unclear. In particular, formally, the notation $\delta_{y(z) \in \Omega}$ does not make sense to me and no explanation is provided.  Since the purpose of the paper is to rigorously derive a precise explosion of the score function, in my opinion, much more details should be added to the current presentation of assumption A2 and its use. The same goes for the use of the Laplace method.

**Resubmission Of Major Revision:**

The authors may consider submitting a major revision at a later time.